# Sea level rise from climate change is expected to increase the release of arsenic into Bangladesh's drinking well water by reduction and by the salt effect

Seth H. Frisbie 1*, Erika J. Mitchell 2, Azizur R. Molla[3]

1 Department of Chemistry and Biochemistry, Norwich University, Northfield, VT, United States of America, 2 Better Life Laboratories, Incorporated, East Calais, VT, United States of America, 3 Department of Public Health, Grand Valley State University, Grand Rapids, MI, United States of America

* sfrisbie@norwich.edu

## Abstract

**Data Availability Statement:** All relevant data are within the paper and its Supporting Information files.

### Background

Over 165,000,000 people live in Bangladesh; approximately 97% of Bangladeshis drink well water. Approximately 49% of Bangladesh's area has drinking well water with arsenic (As) concentrations that exceed the 10 micrograms per liter (μg/L) World Health Organization (WHO) guideline. This exposure to a potent carcinogen is a significant threat to public health. About 21% of Bangladesh is flooded each year during a typical monsoon season. As climate change progresses, sea levels will continue to rise, and the area and duration of these annual floods will increase. We hypothesize that these consequences of climate change can increase the release of arsenic from sediments into Bangladesh's drinking well water.

### Methods

Drinking well water samples were collected during a national-scale survey in Bangladesh. The dissolved oxygen concentration, oxidation-reduction potential, specific conductance, pH, and temperature were measured at sampling with calibrated portable electronic sensors. The arsenic concentration was measured by the silver diethyldithiocarbamate method.

### Results

As the concentration of dissolved oxygen decreases, the concentration of arsenic increases (*p*-value = 0.0028). Relatedly, as the oxidation-reduction potential decreases, the concentration of arsenic increases (*p*-value = 1.3×10$^{-5}$). This suggests that arsenic is released from sediments into Bangladesh's drinking well drinking water by reduction. As the specific conductance increases, the concentration of arsenic increases (*p*-value = 0.023). This suggests that arsenic is also released from sediments into water by the salt effect.

**Funding:** The fieldwork in Bangladesh was funded by the United States Agency of International Development (USAID; contract number US AID RE III 388-0070; https://www.usaid.gov/). This fieldwork began in July of 1997 and ended in August of 1997. USAID is an international development agency that is funded by the United States government. USAID employed Seth H. Frisbie (SHF) and paid his salary during these two months in 1997. After August 1997, SHF received no specific funding for this work. Erika J. Mitchell (EJM) and Azizur R. Molla (ARM) received no specific funding for this work. No commercial companies funded the study or the authors. All other costs have been paid from the personal savings of the authors. The funders had no role in study design, data collection and analysis, decision to publish, or preparation of the manuscript.

**Competing interests:** The authors have declared that no competing interests exist.

## Conclusions

Rising sea levels can cause a decrease in the dissolved oxygen concentration and oxidation-reduction potential of the underlying aquifer; this should increase the dissolution of insoluble arsenate ($H_{3-x}As(V)O_4^{x-}$) in sediments by reduction. This, in turn, should release soluble arsenite ($H_{3-x}As(III)O_3^{x-}$) into the drinking well water. Rising sea levels can cause an increase in the salt concentration of the underlying aquifer; this should increase the release of arsenic from sediments into the drinking well water by the salt effect.

## Introduction

### Overview

In Bangladesh, the intrusion of saltwater caused by sea level rise can increase the salinity of coastal aquifers; this increase in salinity can degrade the drinking water quality [1]. The mixing of sewage with floodwater can also degrade the drinking water quality [2]. These two processes of degrading drinking water quality are well known [1–3]. This paper will discuss two more processes of degrading drinking water quality caused by saltwater intrusion from sea level rise and flooding. These two processes are less well known: the release of arsenic from sediments into drinking well water by reduction and by the salt effect.

Bangladesh, with its low-lying coastal topography, annual monsoons, and frequent cyclones, is expected to be severely impacted by flooding as sea levels continue to rise [4, 5]. The linear relative sea level trend for the Northern Indian Ocean (Bay of Bengal) measured at Visakhapatnam, India, from 1937 to 2013, was 1.05 ± 0.40 millimeters (mm) per year (yr), which is equivalent to a rise of 0.344 ± 0.131 feet in 100 years [6].

In addition, from 1968 to 2012, the water level of the Ganges–Brahmaputra–Meghna River delta in Bangladesh increased by about 3 millimeters per year [7]. At the same time, this increase was slightly faster than the mean sea-level rise at about 2 millimeters per year [7]. This difference in water levels is primarily due to the subsidence of the river delta; from 1993 to 2012, the maximum rates of delta subsidence ranged from 1 millimeter per year to 7 millimeters per year [7]. The principal natural factor contributing to this subsidence is most likely the consolidation of deltaic sediments [7]. The principal human-made factor contributing to this subsidence is most likely the withdrawal of groundwater, especially near Dhaka, the capital of Bangladesh [7].

### Arsenic in Bangladesh's environment

Alluvial sediments deposited in Bangladesh's lowlands during the Holocene epoch (11,700 years ago to the present day) have minerals that often contain arsenic (As) [8, 9]. These minerals release arsenic into Bangladesh's groundwater. This groundwater is used for drinking.

The use of groundwater for drinking has increased tremendously since the 1970s when the government of Bangladesh, United Nations International Children's Emergency Fund (UNICEF), and some non-governmental organizations (NGOs) began installing about 10,000,000 drinking water wells in Bangladesh [10]. Prior to 1970, surface water was the most common source of drinking water in Bangladesh. Approximately 49% of Bangladesh's area contains groundwater with arsenic concentrations greater than the 10 microgram per liter (µg/L) World Health Organization (WHO) drinking water guideline [11, 12]. This new and very large exposure to arsenic in Bangladesh's drinking water is increasing the rates of death and

disease from skin, bladder, liver, and lung cancers and vascular disease; this is a public health crisis [11–13].

## Our research

Our research aims to identify the relationship between the chemical changes of the coastal aquifer due to sea level rise and the increased release of arsenic from sediments into Bangladesh's drinking well water. The effects of dissolved oxygen concentration, oxidation-reduction potential (ORP), and pH on this release of arsenic by reduction are evaluated. In addition, the influences of the salt effect, anion exchange from solid surfaces, and the pairing of ions in water on this release of arsenic by typical ions are also examined. As sea levels continue to rise and floods and cyclones in Bangladesh increase in area and duration, the underlying aquifer can become more saline and more reducing. Our data strongly suggest that these chemical changes can increase the release of arsenic into Bangladesh's drinking well water.

## Materials and methods

### Physical environment

The Peoples' Republic of Bangladesh is located in one of the largest river deltas in the world, the Bengal Delta Plain. The Ganges, Brahmaputra, and Meghna rivers flow through Bangladesh and into the Bay of Bengal (Fig 1) [14]. Most of the country is less than 12 meters (39 feet) above sea level [14]. The average elevation is 8 meters (25 feet) above sea level [4]. In a typical monsoon season about 21% of Bangladesh's land is flooded with a mixture of freshwater from its rivers and saltwater from the Bay of Bengal [4]. The largest recorded flood was during the 1998 monsoon season and covered 68% of Bangladesh's land [4]. The monsoon season is from July to October; in contrast, cyclones are most likely to occur from April to May and from October to November and can rapidly yield over 9 meters (30 feet) of storm surge [4]. These floods and storm surges cause saltwater to intrude from the south to the north (Fig 1) [4, 14].

### Drinking well water sampling and analyses

Groundwater samples were collected from 83 random drinking water wells throughout Bangladesh from July 22 to August 14, 1997 (Fig 1). These wells were generally hand-pumped wells, locally called "tubewells," and were screened at 4.6 meters to 370 meters (15 feet to 1,200 feet) below ground surface (bgs). Stratified random sampling was used to select the general location of each drinking water well. More specifically, a map of Bangladesh was divided into grids and one drinking water well was sampled from each grid. These samples were collected by traveling over roads, paths, and rivers, stopping at the preselected location in the grid, finding the nearest drinking water well, and sampling this well. The latitude and longitude of all sample locations were identified by using the Global Positioning System (GPS). The depth of the drinking water well was reported by the owner or a principal user at sampling.

Established collection, preservation, and storage methodologies were used to ensure that each sample was representative of groundwater quality [16, 17]. Accordingly, all sampled drinking water wells were purged by vigorous pumping for 10 minutes immediately before analysis in the field and sample collection for analysis in the laboratory. Each sample was split into two subsamples: one for analyses in the field and one for analysis at the laboratory.

Subsamples were analyzed in the field immediately after collection for dissolved oxygen concentration, oxidation-reduction potential, specific conductance (SC), pH, and temperature (T) with calibrated portable electronic sensors. Dissolved oxygen was measured with a

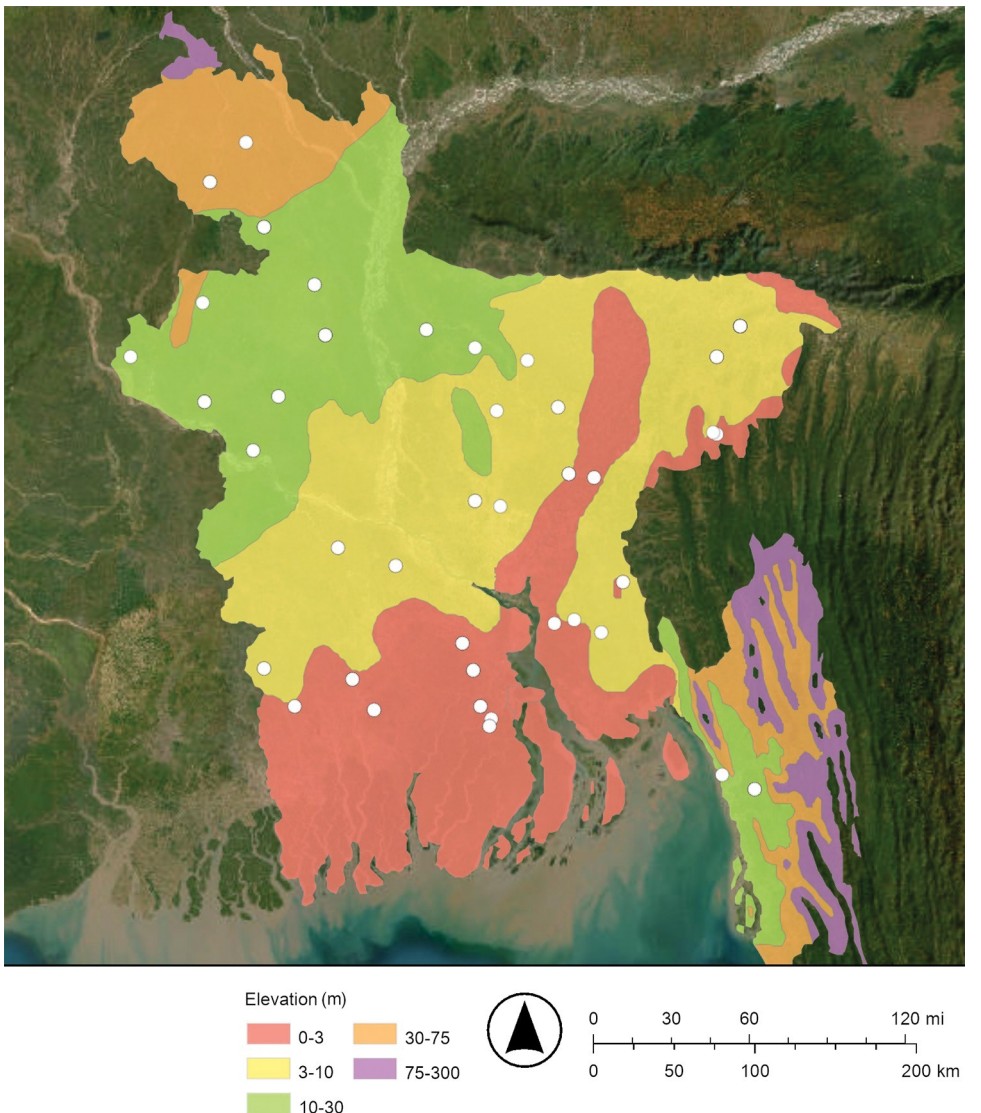

**Fig 1. Map of Bangladesh showing the locations where groundwater samples were collected from drinking water wells.** The Ganges (Padma), Bramaputra, Jamuna, and Meghana Rivers flow south into the Bay of Bengal and the Indian Ocean. The map was created with ArcGIS® Online using a public domain basemap published by the United States Geological Survey [15].

Hanna® HI 9143 meter. Oxidation-reduction potential, pH, and temperature were measured with Cole-Parmer® Digi-Sense meters. Specific conductance was measured with a Cole-Parmer® Model 19815 meter.

Subsamples were also collected for analysis in the laboratory directly into 350 milliliter (mL) polyethylene bottles and were not filtered. These subsamples were preserved by acidification to pH < 2 with 18.6% (weight/weight) nitric acid ($HNO_3$) and stored in ice-packed coolers for transportation to the laboratory. The temperature of all laboratory subsamples was maintained at 0 Celsius (°C) to 4°C until immediately before analysis.

The concentration of arsenic was measured at the International Centre for Diarrhoeal Disease Research, Bangladesh (ICDDR,B) laboratory by the silver diethyldithiocarbamate method [16].

## Statistical analyses

Statistical analyses were performed using R version 4.2.2 (2022-10-31 ucrt) "Innocent and Trusting." These analyses were confirmed using Microsoft® Excel® for Microsoft 365, version 2302, build 16.0.16130.20298. Linear regression was used to determine the effect of the dependent variable arsenic (As) in micrograms per liter (μg/L) on the independent variables dissolved oxygen in milligrams per liter (mg/L), oxidation-reduction potential in millivolts (mV), specific conductance in microsiemens per centimeter (μS/cm), pH, and temperature in Celsius (°C). Each of these five regressions were tested at the 95% confidence level. Then these five test results were corrected for the expected proportion of falsely rejected null hypotheses from multiple comparisons by using Benjamini and Hochberg's False Discovery Rate [18]. According to the standard Benjamini and Hochberg methodology, the five calculated $p$-values were arranged from smallest to largest, then compared to their rank divided by the total number of comparisons and multiplied this value by the alpha (α) = 0.05 significance threshold [18].

## Mapping

Maps of sample location and arsenic concentration, dissolved oxygen concentration, oxidation-reduction potential, specific conductance, pH, and temperature of Bangladesh's drinking well water were created using ArcGIS® Online software version 2023.3 by Esri®. ArcGIS® Online is the intellectual property of Esri® and is used herein under license. Copyright Esri®. All rights reserved. For more information about Esri® software, please visit www.esri.com.

## Inclusivity in global research

Additional information regarding the ethical, cultural, and scientific considerations specific to inclusivity in global research is included in the S1 Checklist. This work was done at the request of the Government of Bangladesh by the United States Agency of International Development (USAID). More specifically, it was completed under the direction of the Bangladesh Rural Electrification Board. This study did not use animal or human subjects; therefore, the Government of Bangladesh did not require permits or approval from an ethics board.

# Results and discussion

## Descriptive statistics

The minimum, first quartile, median or second quartile, sample mean, third quartile, maximum, sample standard deviation ($s$), and number of observations ($n$) for the laboratory analysis of arsenic concentration and the field measurements at sampling of dissolved oxygen, oxidation-reduction potential, specific conductance, pH, and temperature are listed in Table 1. The raw data and statistical analyses are included in the S1 and S2 Files.

## Correcting for Benjamini and Hochberg's False Discovery Rate

If 100 comparisons are made at α = 0.05 and the null hypothesis is always true, then in the long run 95% of the tests will correctly accept the null hypothesis and 5% of the tests will incorrectly reject the null hypothesis. This incorrect rejection of a true null hypothesis, also called a type I error or a false positive, from multiple comparisons was addressed by using Benjamini and Hochberg's False Discovery Rate as follows [18].

Our data set uses five comparisons (Table 2). The Benjamini and Hochberg method sequentially increases the threshold of significance for tests with multiple comparisons to reduce the incidence of false positives [18]. More specifically, the $p$-values for the five

**Table 1. The descriptive statistics for arsenic (As) in micrograms per liter (μg/L), dissolved oxygen (DO; O$_{2(aq)}$) in milligrams per liter (mg/L), oxidation-reduction potential (ORP) in millivolts (mV), specific conductance (SC) in microsiemens per centimeter (μS/cm), pH, and temperature (T) in Celsius (˚C).**

| Statistic | As (μg/L) | DO (mg/L) | ORP (mV) | SC (μS/cm) | pH | T (˚C) |
|---|---|---|---|---|---|---|
| Minimum | 0.0 | 0.16 | −120 | 40 | 3.90 | 24.3 |
| 1$^{st}$ Quartile | 6.3 | 2.54 | −71 | 163 | 6.16 | 26.1 |
| Median | 32 | 3.57 | −12 | 494 | 6.70 | 26.6 |
| Mean | 77 | 4.25 | 19 | 644 | 6.57 | 26.9 |
| 3$^{rd}$ Quartile | 80 | 5.43 | 103 | 798 | 6.93 | 27.4 |
| Maximum | 450 | 12.70 | 276 | 5,360 | 7.96 | 32.7 |
| Sample Standard Deviation (s) | 110 | 2.50 | 109 | 747 | 0.70 | 1.5 |
| Number of Observations (n) | 83 | 82 | 83 | 82 | 83 | 83 |

regressions in this study listed from smallest to largest are $1.3{\times}10^{-5}$, 0.0028, 0.023, 0.032, and 0.26 (Table 2). Since the smallest p-value, $1.3{\times}10^{-5}$, is less than or equal to $p_{(1)}$ = (1/5) 0.05 = 0.01 the effect of arsenic concentration on oxidation-reduction potential is statistically significant at a Benjamini and Hochberg's False Discovery Rate corrected 95% confidence level. Since the second smallest p-value, 0.0028, is less than or equal to $p_{(2)}$ = (2/5)0.05 = 0.02 the effect of arsenic concentration on dissolved oxygen is statistically significant at a corrected 95% confidence level. Since the third smallest p-value, 0.023, is less than or equal to $p_{(3)}$ = (3/5) 0.05 = 0.03 the effect of arsenic concentration on specific conductance is statistically significant at a corrected 95% confidence level. Since the fourth smallest p-value, 0.032, is less than or equal to $p_{(4)}$ = (4/5)0.05 = 0.04 the effect of arsenic concentration on pH is statistically significant at a corrected 95% confidence level. And since the largest p-value, 0.26, is greater than $p_{(5)}$ = (5/5)0.05 = 0.05 the effect of arsenic concentration on temperature is not statistically significant at a corrected 95% confidence level. Therefore, the conclusions from each of our five regressions are unchanged after correcting for the expected proportion of falsely rejected null hypotheses from multiple comparisons (Table 2) [18].

## The release of arsenic into Bangladesh's drinking well water by reduction, and the expected impact of sea level rise

**Background.** The arsenic containing minerals that are contaminating Bangladesh's groundwater are a mixture of iron (Fe) oxyhydroxides and manganese (Mn) oxyhydroxides [8, 19]. Arsenic is released from these minerals into drinking well water by reduction. More specifically, the reduction of insoluble arsenate ($H_{3-x}As(V)O_4^{x-}$) to soluble arsenite ($H_{3-x}As$

**Table 2. Using Benjamini and Hochberg's False Discovery Rate to correct for the expected proportion of falsely rejected null hypotheses from the five comparisons of arsenic (As) in micrograms per liter (μg/L) with dissolved oxygen (DO; O$_{2(aq)}$) in milligrams per liter (mg/L), oxidation-reduction potential (ORP) in millivolts (mV), specific conductance (SC) in microsiemens per centimeter (μS/cm), pH, and temperature (T) in Celsius (˚C) [18].**

| Regression Equation | p-value | $p_{(i)} = \left(\frac{i}{5}\right)0.05$ | Conclusion |
|---|---|---|---|
| $As\left(\frac{\mu g}{L}\right) = 85.3 - 0.454(ORP(mV))$ | $1.3{\times}10^{-5}$ | 0.01 | Significant [a] |
| $As\left(\frac{\mu g}{L}\right) = 137 - 14.1\left(DO\left(\frac{mg}{L}\right)\right)$ | 0.0028 | 0.02 | Significant [a] |
| $As\left(\frac{\mu g}{L}\right) = 53.8 + 0.0364\left(SC\left(\frac{\mu S}{cm}\right)\right)$ | 0.023 | 0.03 | Significant [a] |
| $As\left(\frac{\mu g}{L}\right) = -162 + 36.3(pH)$ | 0.032 | 0.04 | Significant [a] |
| $As\left(\frac{\mu g}{L}\right) = -173 + 9.31(T(˚C))$ | 0.26 | 0.05 | Not significant [b] |

[a] Statistically significant at a Benjamini and Hochberg's False Discovery Rate corrected 95% confidence level.
[b] Not statistically significant at a Benjamini and Hochberg's False Discovery Rate corrected 95% confidence level.

(III)$O_3^{x-}$) releases arsenic from these minerals into Bangladesh's drinking well water [8, 19, 20]. In addition, the dissolution of the iron oxyhydroxides by the reduction of insoluble Fe(III) to soluble Fe(II) may also release arsenic into Bangladesh's drinking well water. Similarly, the dissolution of the manganese oxyhydroxides by the reduction of insoluble Mn(IV) or insoluble Mn(III) to soluble Mn(II) may also release arsenic into Bangladesh's drinking well water [21].

This reducing environment is often enhanced by seasonal flooding from the annual monsoons and cyclones, and by the cultivation of rice in flooded paddies. These floodwaters are a barrier to the diffusion of oxygen ($O_{2(g)}$), a very strong oxidizing agent, from the atmosphere into the aquifer [8, 19, 20]. During flooding, heterotrophic microorganisms can deplete the concentration of dissolved oxygen (DO; $O_{2(aq)}$) in groundwater; this also helps to make a reducing environment [8, 21, 22]. (The phase designations "g" and "aq" denote "gas" and "aqueous," respectively).

An oxidation-reduction (redox) reaction is the transfer of one or more electrons from an atom or molecule to another atom or molecule. An oxidation reaction produces an electron or electrons ($e^-$) as follows (Reaction 1) [23]. (The phase designations "aq," "g," and "l," denote "aqueous," "gas," and "liquid," respectively).

$$H_3As(III)O_{3(aq)} + H_2O_{(l)} \rightleftharpoons H_3As(V)O_{4(aq)} + 2H^+_{(aq)} + 2e^- \quad E^0_{ox} = -0.560 \text{ V} \qquad (1)$$

The arsenic atom (As) in arsenous acid ($H_3As(III)O_3$) is in the +3 or As(III) oxidation state. The arsenic atom (As) in arsenic acid ($H_3As(V)O_4$) is in the +5 or As(V) oxidation state. Therefore, the As(III) atom is reduced relative to the As(V) atom, which is oxidized. Oxidation states are useful for tracking the transfer of electrons during a redox reaction.

A reduction reaction consumes an electron or electrons as follows (Reaction 2) [23].

$$O_{2(g)} + 4H^+_{(aq)} + 4e^- \rightleftharpoons 2H_2O_{(l)} \quad E^0_{red} = 1.229 \text{ V} \qquad (2)$$

The oxygen atoms (O) in molecular oxygen ($O_{2(g)}$) are in the 0 oxidation state. The oxygen atom (O) in a water molecule ($H_2O_{(l)}$) is in the −2 oxidation state. Therefore, the O atoms in $O_{2(g)}$ are oxidized relative to the O atom in $H_2O_{(l)}$, which is reduced.

The combined oxidation-reduction reaction follows (Reaction 3).

$$2H_3As(III)O_{3(aq)} + O_{2(g)} \rightleftharpoons 2H_3As(V)O_{4(aq)} \quad E^0_{cell} = 0.669 \text{ V} \qquad (3)$$

Arsenous acid ($H_3As(III)O_3$) is the reducing agent; it causes the reduction of $O_{2(g)}$. Molecular oxygen ($O_{2(g)}$) is the oxidizing agent; it causes the oxidation of $H_3As(III)O_3$. The standard electrode potential ($E^0_{cell}$) for this oxidation-reduction reaction is
$E^0_{cell} = E^0_{ox} + E^0_{red} = -0.560 \text{ V} + 1.229 \text{ V} = 0.669 \text{ V}$. This positive $E^0_{cell}$ means that the reaction is thermodynamically favored at standard conditions; that is, the reactants are expected to spontaneously make products when all gases are at 1.0 atmosphere (atm) pressure and all solutes are at 1.0 molar (M) concentration [24]. The Nernst equation gives the electrode potential of an oxidation-reduction reaction at nonstandard conditions [24].

This spontaneous oxidation of As(III) to As(V) by molecular oxygen is very important. First, the minimum voltage that is required for the electrolysis of liquid water ($H_2O_{(l)}$) to oxygen gas ($O_{2(g)}$) and hydrogen gas ($H_{2(g)}$) at standard conditions is 1.229 volts; therefore, dissolved oxygen is one of the most powerful oxidizing agents that can exist in liquid water (Reaction 2) [17]. Second, As(V) is relatively insoluble and As(III) is relatively soluble; therefore, arsenate ($H_{3-x}As(V)O_4^{x-}$) is expected to be in solid sediments, and arsenite ($H_{3-x}As(III)O_3^{x-}$) is expected to be dissolved in drinking well water [8, 19, 20].

**Dissolved oxygen and the release of arsenic.** As sea levels continue to rise and floods and cyclones in Bangladesh increase in area and duration, the underlying aquifer can become

more reducing. These floodwaters are a barrier to the diffusion of oxygen ($O_{2(g)}$), a very strong oxidizing agent, from the atmosphere into the aquifer [8, 19, 20]. During flooding, heterotrophic microorganisms can deplete the concentration of dissolved oxygen (DO; $O_{2(aq)}$) in the floodwater and in the groundwater; this also helps to make a reducing environment [8, 21, 22]. The following data strongly suggest that this chemical change can increase the release of arsenic into Bangladesh's drinking well water by reduction.

Again, dissolved oxygen is one of the most powerful oxidizing agents that can exist in liquid water (Reaction 2) [17]. If the dissolved oxygen concentration is relatively large, then the oxidation of soluble arsenite ($H_{3-x}As(III)O_3^{x-}$) to insoluble arsenate ($H_{3-x}As(V)O_4^{x-}$) is favored (Reaction 3). That is, if the system is oxidizing; this equilibrium is shifted to the right, and the arsenic precipitates and is removed from the drinking well water. If the dissolved oxygen concentration is relatively small, then the reduction of insoluble arsenate ($H_{3-x}As(V)O_4^{x-}$) to soluble arsenite ($H_{3-x}As(III)O_3^{x-}$) is favored (Reaction 3). That is, if the system is reducing; this equilibrium is shifted to the left, and the arsenic dissolves and is added to the drinking well water.

Maps of the concentrations of arsenic and dissolved oxygen in Bangladesh's drinking well water are shown in Figs 2 and 3, respectively. These maps suggest that when the concentration of dissolved oxygen decreases, the concentration of arsenic increases. The dissolved oxygen concentration of drinking well water in this study ranged from 0.16 mg/L to 12.70 mg/L (Fig 3; Table 1). As predicted, relatively low concentrations of dissolved oxygen are associated with relatively high concentrations of arsenic. The map of the dissolved oxygen concentration of Bangladesh's drinking well water has lower dissolved oxygen concentrations in the south near the Bay of Bengal and higher dissolved oxygen concentrations in the north near the Himalayan mountains; this observation is consistent with floods and storm surges intruding from the south to the north (Fig 3).

The scatterplot, linear regression equation, and $p$-value for the concentration of arsenic versus the concentration of dissolved oxygen are shown in Fig 4. This linear regression gives a statistically significant negative slope (−14.1 μg/mg; Fig 4; Table 2). This suggests that arsenic is released from sediments into Bangladesh's drinking well water by reduction. More specifically, this inverse relationship between arsenic concentration and dissolved oxygen concentration suggests that in the absence of dissolved oxygen, insoluble arsenate ($H_{3-x}As(V)O_4^{x-}$) from sediments is reduced to soluble arsenite ($H_{3-x}As(III)O_3^{x-}$) and released into Bangladesh's drinking well water (Reaction 3) [8, 20].

The regression in Fig 4 describes the increase in arsenic concentration when a specific, common, and powerful oxidizing agent, dissolved oxygen, is relatively absent (Fig 4). The relative absence of dissolved oxygen and the consequent release of arsenic into drinking well water is very likely caused by the seasonal and episodic flooding in the Bengal Delta Plain (Bangladesh and Northeast India), as suggested by the maps of dissolved oxygen concentration and arsenic concentration (Figs 2 and 3) [8, 20, 25].

More specifically, about 21% of Bangladesh is flooded each year during a typical monsoon season (Fig 1) [4, 14]. These floodwaters restrict the diffusion oxygen ($O_{2(g)}$) from the atmosphere into groundwater; this makes a reducing environment [8, 21, 22]. The reduction of insoluble arsenate ($H_{3-x}As(V)O_4^{x-}$) to soluble arsenite ($H_{3-x}As(III)O_3^{x-}$) releases arsenic from iron oxyhydroxide and manganese oxyhydroxide minerals into Bangladesh's drinking well water [8, 19, 20]. In addition, the dissolution of the iron oxyhydroxides by the reduction of insoluble Fe(III) to soluble Fe(II), and the dissolution of the manganese oxyhydroxides by the reduction of insoluble Mn(IV) or insoluble Mn(III) to soluble Mn(II) may also release arsenic into Bangladesh's drinking well water [21]. That is, dissolving these bulk minerals may also release arsenic into Bangladesh's drinking well water. In conclusion, as the climate

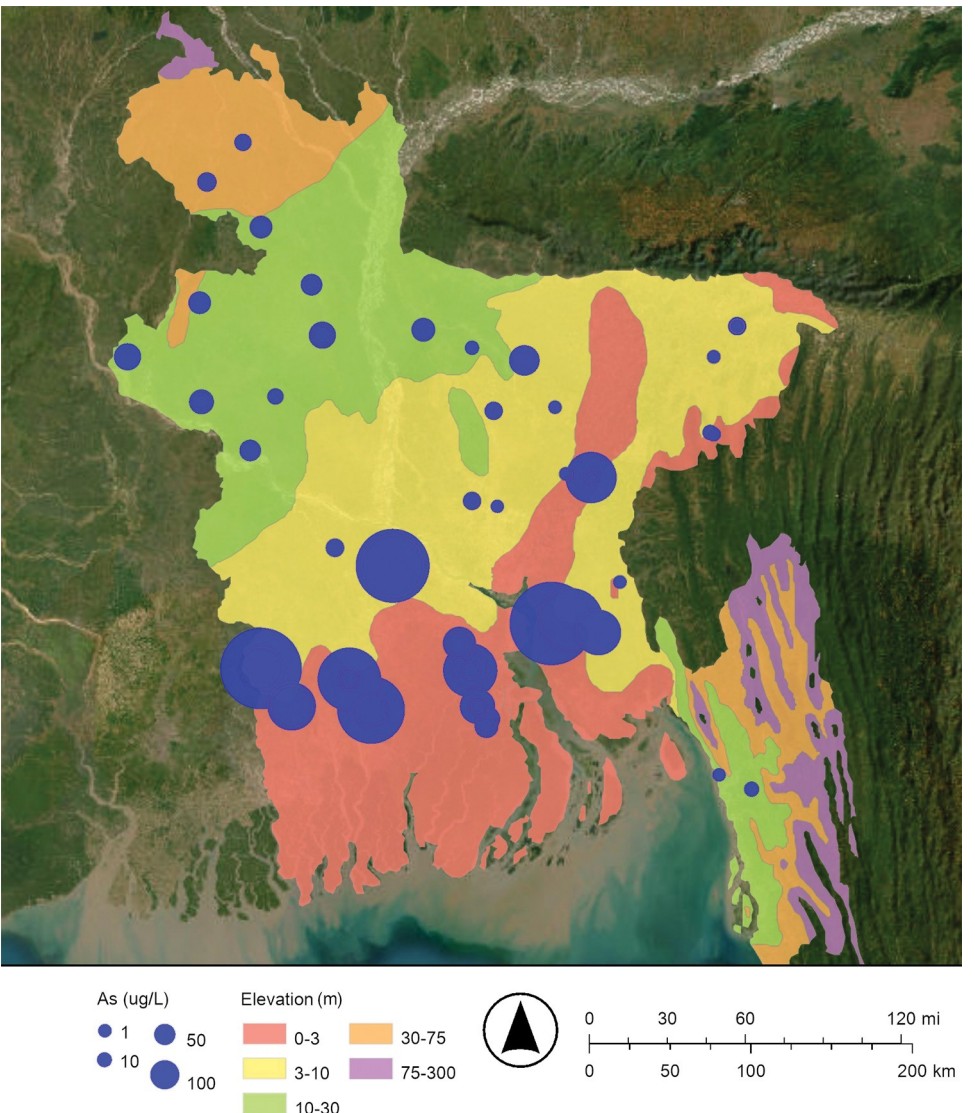

**Fig 2. Proportional symbol map of arsenic (As) in micrograms per liter (μg/L) in Bangladesh's drinking well water.** The World Health Organization (WHO) drinking water guideline for arsenic is 10 μg/L, and the Bangladesh national drinking water standard for arsenic is 50 μg/L [12]. The map was created with ArcGIS® Online using a public domain basemap published by the United States Geological Survey [15].

changes and sea levels rise and flooding increases, the release of arsenic into Bangladesh's drinking well water by reduction is expected to increase.

**Oxidation-reduction potential and the release of arsenic.**   Dissolved oxygen is a single oxidizing agent in groundwater; in contrast, oxidation-reduction potential is the measurement of all oxidizing and all reducing agents in groundwater. If the oxygen-potential is relatively high, then the oxidation of soluble arsenite ($H_{3-x}As(III)O_3^{x-}$) to insoluble arsenate ($H_{3-x}As(V)O_4^{x-}$) is favored. If the oxygen-potential is relatively low, then the reduction of insoluble arsenate ($H_{3-x}As(V)O_4^{x-}$) to soluble arsenite ($H_{3-x}As(III)O_3^{x-}$) is favored.

Maps of the concentration of arsenic and the oxidation-reduction potential of Bangladesh's drinking well water are shown in Figs 2 and 5, respectively. These maps suggest that when the oxidation-reduction potential decreases, the concentration of arsenic increases. The

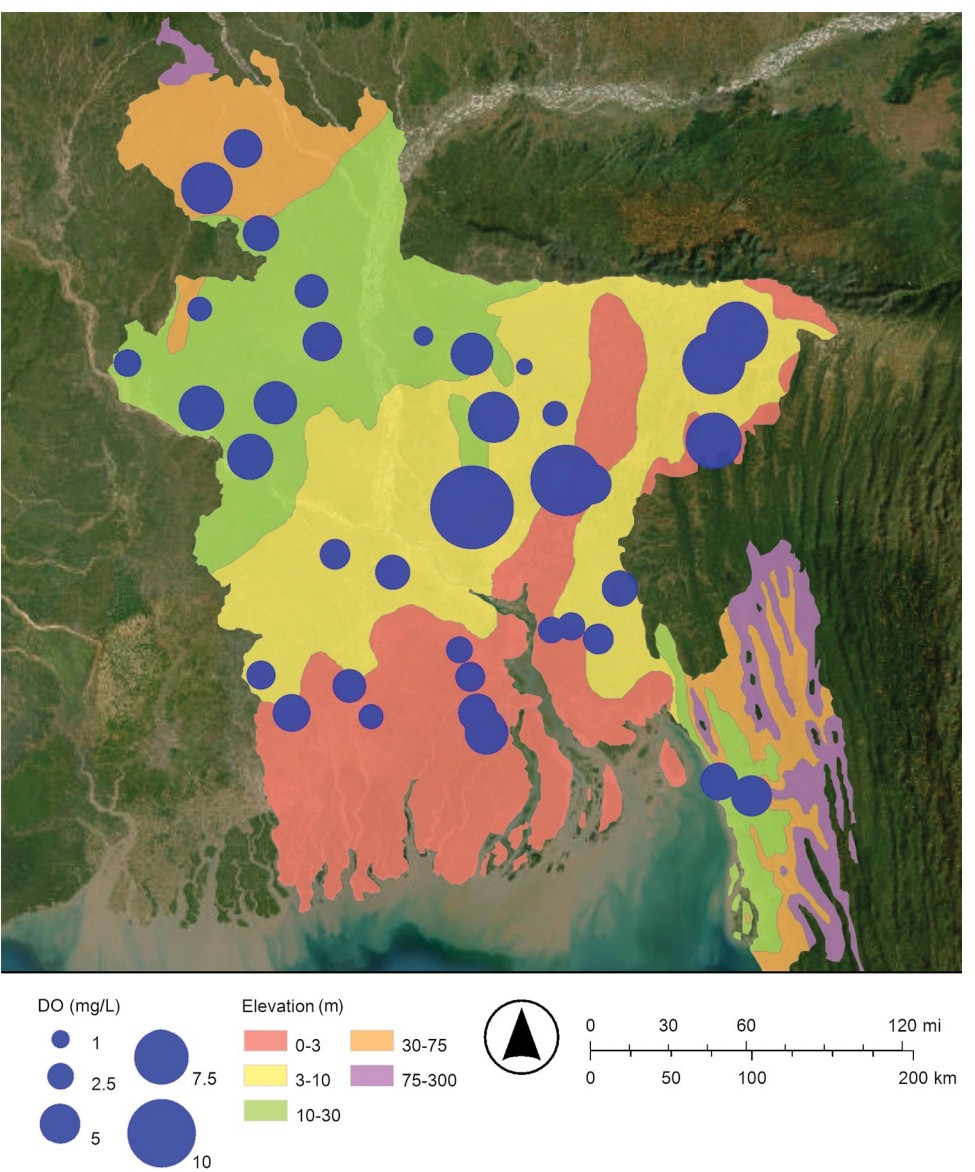

**Fig 3. Proportional symbol map of dissolved oxygen (DO) in milligrams per liter (mg/L) in Bangladesh's drinking well water.** The map was created with ArcGIS® Online using a public domain basemap published by the United States Geological Survey [15].

oxidation-reduction potential of drinking well water in this study ranged from −120 mV to 276 mV (Fig 5; Table 1). The map of the oxidation-reduction potential of Bangladesh's drinking well has lower oxidation-reduction potentials in the south near the Bay of Bengal and higher oxidation-reduction potentials in the north near the Himalayan mountains; this is consistent with floods and storm surges intruding from the south to the north (Fig 5).

The scatterplot, linear regression equation, and $p$-value for the concentration of arsenic versus oxidation-reduction potential are shown in Fig 6. Similarly, this linear regression gives a statistically significant negative slope (−0.454 μg/L mV, Fig 6; Table 2). This also suggests that arsenic is released from sediments into Bangladesh's drinking well water by reduction. More specifically, this inverse relationship between arsenic concentration and oxidation-reduction potential suggests that insoluble arsenate ($H_{3-x}As(V)O_4^{x-}$) from sediments in a reducing

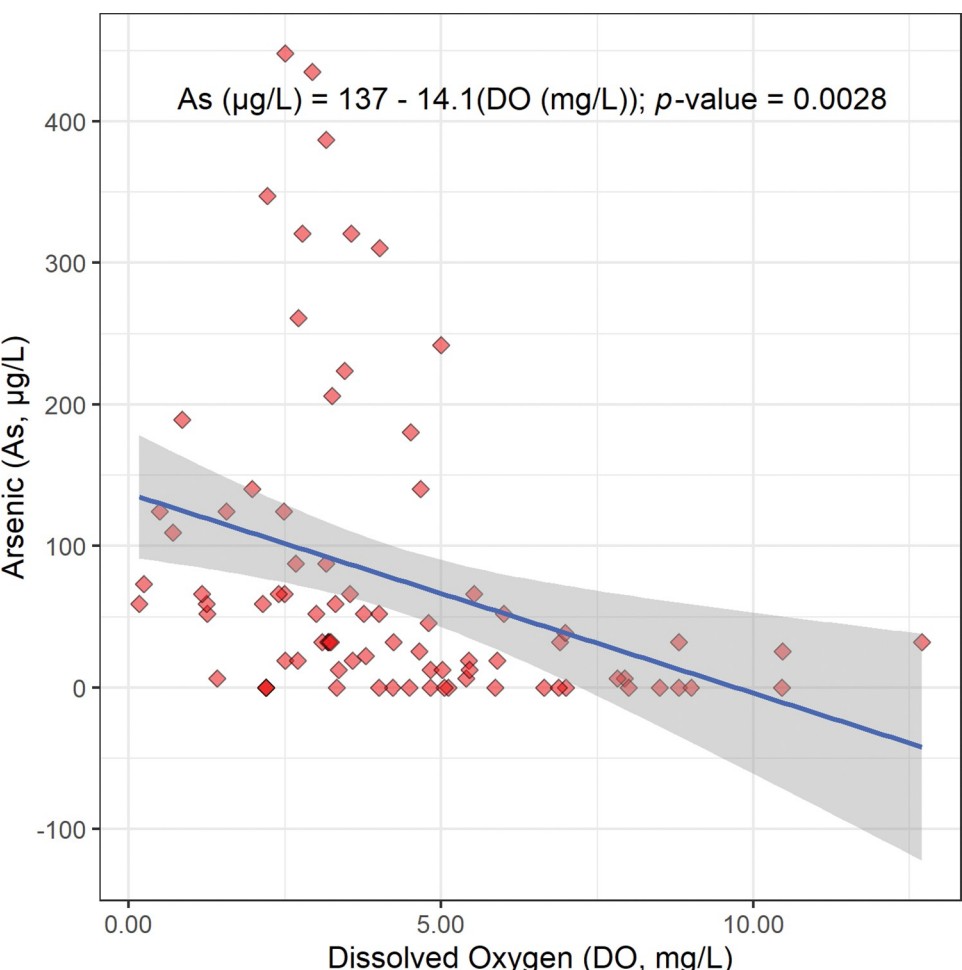

As (μg/L) = 137 - 14.1(DO (mg/L)); *p*-value = 0.0028

**Fig 4. The scatterplot, linear regression equation, 95% confidence band, and *p*-value for the concentration of arsenic (As) in micrograms per liter (μg/L) versus the concentration of dissolved oxygen (DO) in milligrams per liter (mg/L).** This regression is statistically significant at a Benjamini and Hochberg's False Discovery Rate corrected 95% confidence level (Table 2) [18].

environment is reduced to soluble arsenite ($H_{3-x}As(III)O_3^{x-}$) and released into Bangladesh's drinking well water (Reaction 3) [8, 20].

The regression in Fig 6 describes the increase in arsenic concentration in a reducing environment (Fig 6). This reducing environment and release of arsenic into drinking well water is very likely caused by the seasonal flooding in Bangladesh, as suggested by the maps of oxidation-reduction potential and arsenic concentration (Figs 2 and 5) [8, 20, 25]. In conclusion, as the climate changes and sea levels rise and flooding increases, the release of arsenic into Bangladesh's drinking well water by reduction is expected to increase.

**The Nernst equation and the release of arsenic.** The Nernst equation gives the electrode potential of an oxidation-reduction reaction at nonstandard conditions [24]. That is, the Nernst equation gives the relative voltage when the reactants and products are not at 1.0 molar for solutes and are not at 1.0 atmospheres for gases [24]. In this study, the maximum concentration of arsenic is less than 1.0 M; it is $5.98 \times 10^{-6}$ M (448 μg/L); therefore, the oxidation or reduction of arsenic in Bangladesh's drinking well water is always at nonstandard conditions and is best described by the Nernst equation.

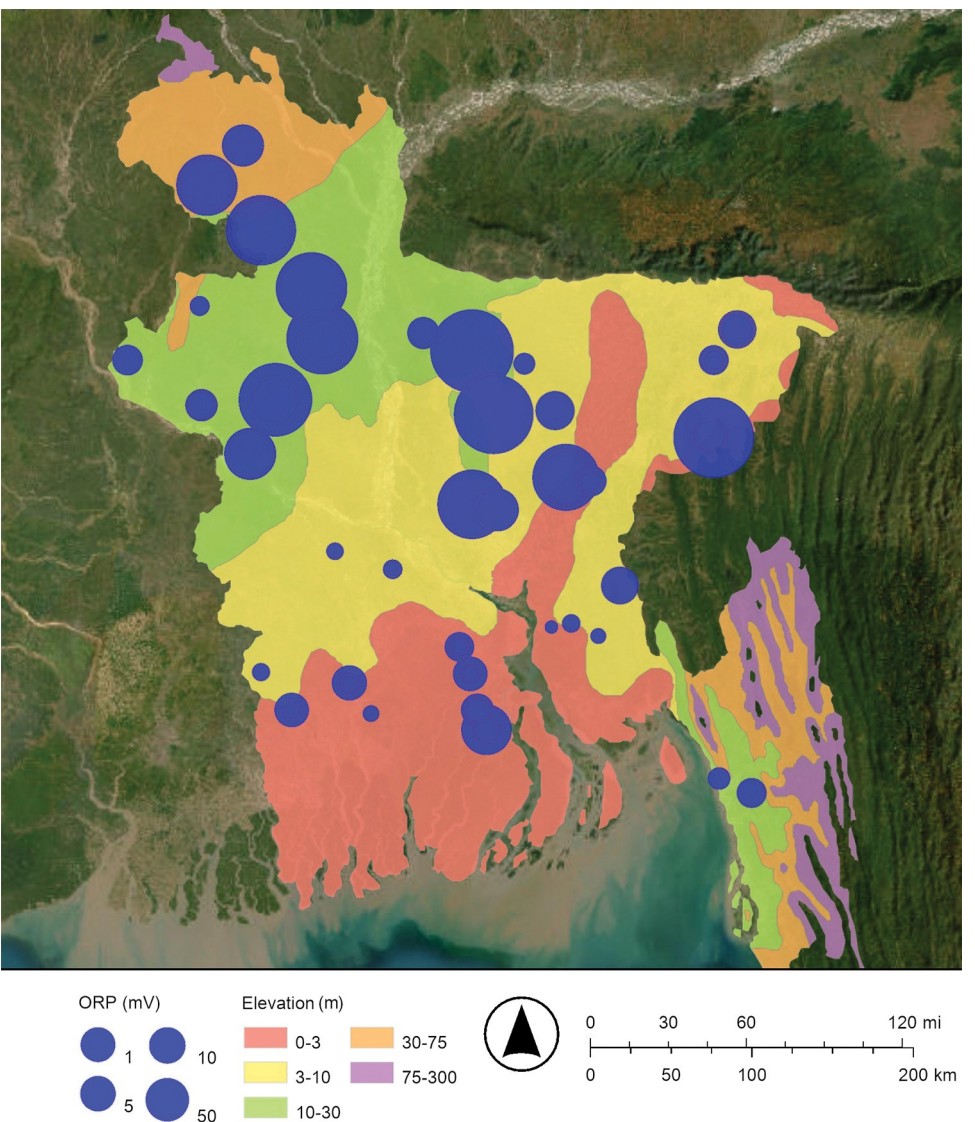

**Fig 5. Proportional symbol map of oxidation-reduction potential (ORP) in millivolts (mV) of Bangladesh's drinking well water.** The map was created with ArcGIS® Online using a public domain basemap published by the United States Geological Survey [15].

The scatterplot, linear regression equation, and *p*-value for oxidation-reduction potential versus pH are shown in Fig 7.

This regression equation is an estimate of an overall Nernst equation for Bangladesh's aquifer (Fig 7). If the oxidation-reduction environment of this system were controlled by a single redox reaction, then the scattering around this regression line would be relatively small. However, this scattering is relatively large, suggesting that a mixture of different minerals, such as iron oxyhydroxides, manganese oxyhydroxides, and possibly other minerals, are controlling the redox environment of Bangladesh's aquifer [8, 19]. It also suggests that a mixture of different solutes and gases at different concentrations and pressures are also controlling this redox environment.

In addition, this linear regression of oxidation-reduction potential versus pH gives a statistically significant negative slope (−70.8 mV; Fig 7). This suggests that the dominant redox

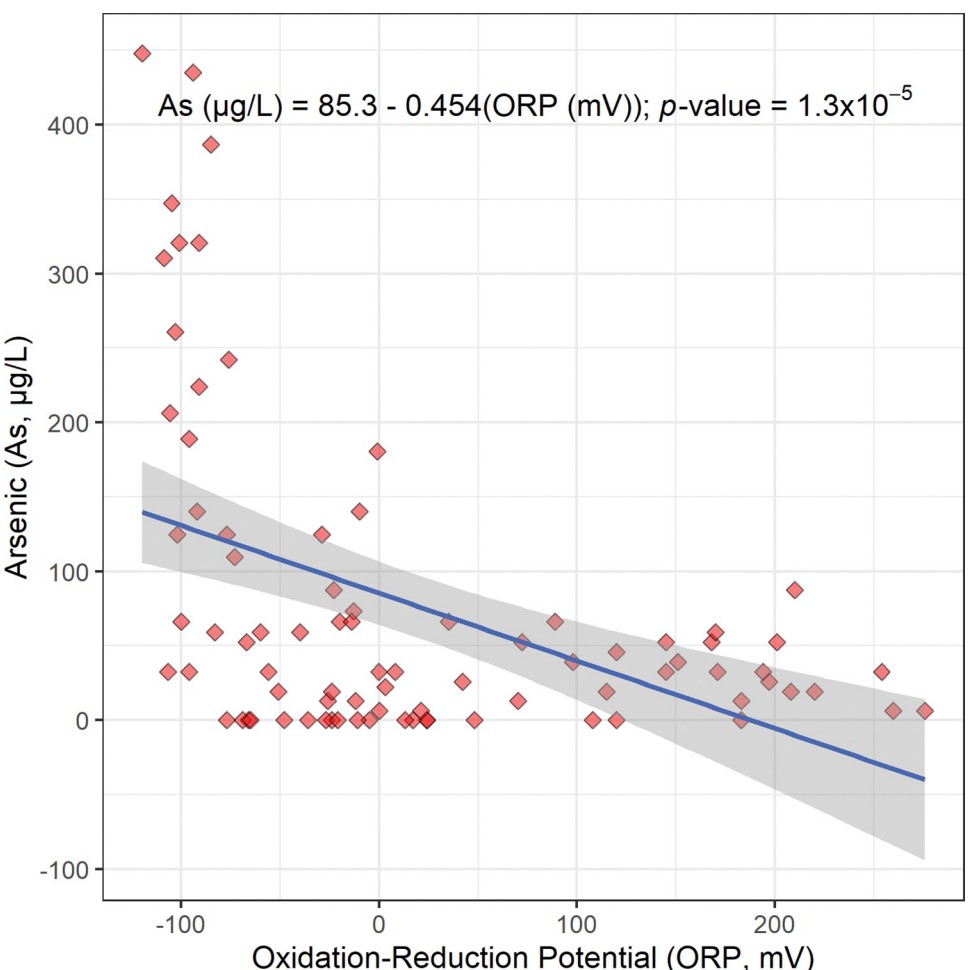

**Fig 6. The scatterplot, linear regression equation, 95% confidence band, and *p*-value for the concentration of arsenic (As) in micrograms per liter (µg/L) versus oxidation-reduction potential (ORP) in millivolts (mV).** This regression is statistically significant at a Benjamini and Hochberg's False Discovery Rate corrected 95% confidence level (Table 2) [18].

reaction or reactions in Bangladesh's aquifer is pH dependent, and more specifically, that aqueous hydrogen ion ($H^+_{(aq)}$) is a reactant in this dominant redox reaction or reactions as written. This gives the following general redox reaction and overall Nernst equation for this system; a moles of A and m moles of $H^+_{(aq)}$ react to make b moles of B as follows (Reaction 4) [17]. The derivation and application of a pH-dependent Nernst equation are included in the S1 and S3 Files.

$$aA + mH^+ \rightleftharpoons bB \tag{4}$$

$$E_{cell} = E^\circ_{cell} - \frac{2.303RT}{nF}\log_{10}\left(\frac{[B]^b}{[A]^a[H^+]^m}\right) = E^\circ_{cell} - \frac{2.303RT}{nF}\log_{10}\left(\frac{[B]^b}{[A]^a}\right) - \frac{2.303RTm}{nF}pH$$

In contrast, the oxidation of As(III) by oxygen to As(V) is not pH dependent (Reaction 3); it does not use aqueous hydrogen ion ($H^+_{(aq)}$) or aqueous hydroxide ion ($OH^-_{(aq)}$) as reactants or as products. Therefore, the oxidation of As(III) to As(V) and the reduction of As(V) to As (III) are not controlling the redox environment of Bangladesh's aquifer (Reaction 3). This lack

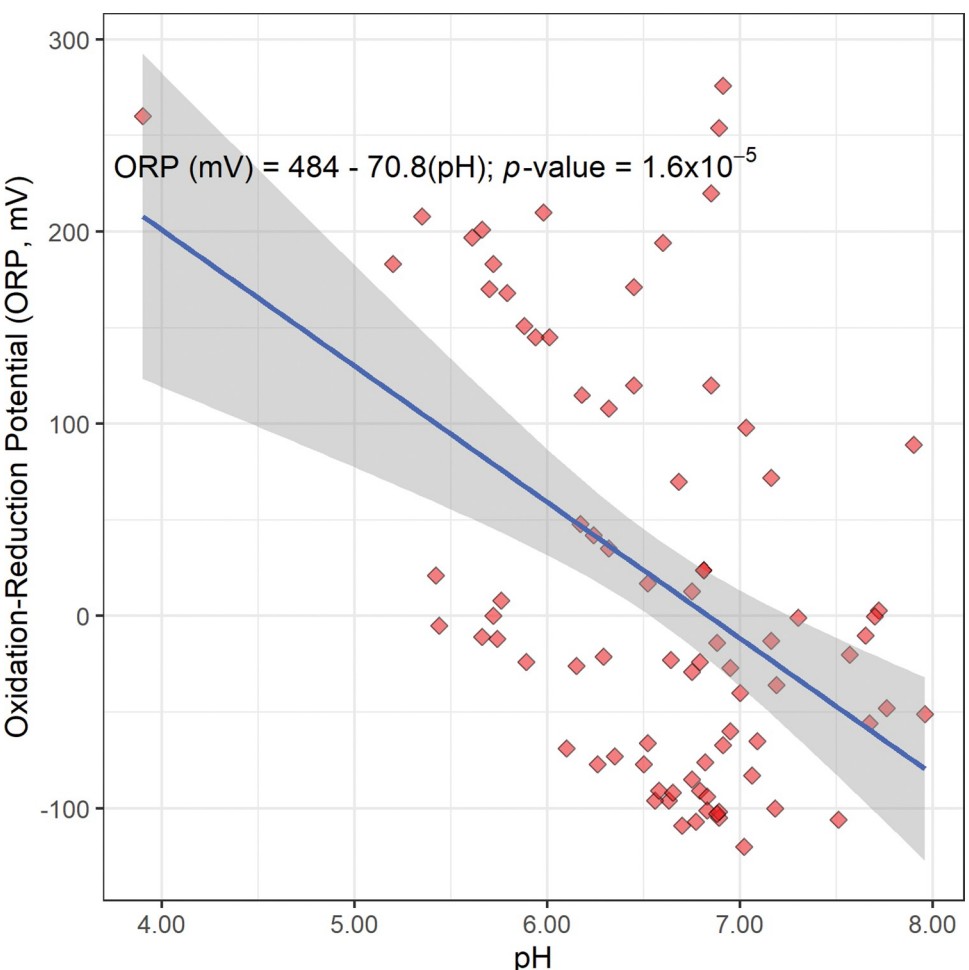

ORP (mV) = 484 - 70.8(pH); *p*-value = 1.6x10$^{-5}$

**Fig 7. The scatterplot, linear regression equation, 95% confidence band, and *p*-value for the oxidation-reduction potential (ORP) in millivolts (mV) versus pH.** This regression is statistically significant at the 95% confidence level.

of control by Reaction 3 is most likely caused by the relative abundances of iron, manganese, and arsenic in Bangladesh's aquifer. Iron, manganese, and arsenic can exist in more than one oxidation state in water and, as a result, can participate in redox reactions in aquifers. Globally, iron at 62,000 milligrams per kilogram (mg/kg) is the 4[th] most abundant element in the earth's crust, manganese at 1,060 mg/kg is the 12[th], and arsenic at 1.8 mg/kg is the 51[st] [26]. These relative abundances suggest that iron and manganese are more likely to control the redox environment of an aquifer than arsenic; this supports the observation that the redox environment of Bangladesh's aquifer is pH dependent and, as a result, is not controlled by arsenic.

In conclusion, the relatively large scattering around the regression line in Fig 7, the difference in pH dependence between Reactions 3 and 4, and the dominance of iron and manganese relative to arsenic strongly suggests that the redox environment of Bangladesh's aquifer is controlled by a mixture of iron oxyhydroxides, manganese oxyhydroxides, and possibly other minerals. Conversely, these observations strongly suggests that the redox environment of Bangladesh's aquifer is not controlled by a single chemical reaction. More specifically, these observations strongly suggests that the redox environment of Bangladesh's aquifer is not controlled by Reaction 3.

Finally, this dominant redox reaction at equilibrium and at a constant temperature has a y-intercept that equals $E_{cell}^{\circ} - \frac{2.303RT}{nF} \log_{10}\left(\frac{[B]^b}{[A]^a}\right)$ and a slope that equals $-\frac{2.303RTm}{nF}$ (Reaction 4). $\frac{2.303RT}{F} = 59.5$ mV at R = 8.314462618 J K$^{-1}$ mol$^{-1}$, T = 300.0 kelvin (K; 26.9°C; the average temperature of groundwater in this study; Table 1), 1 volt (Joule/coulomb; J/C) = 1,000 milli-volts, and F = 96485.33212 C mol$^{-1}$ (S1 File); therefore, the 95% confidence interval for $\frac{m}{n}$, the moles of aqueous hydrogen ion reacted (H$^+_{(aq)}$) over the moles of electrons transferred (e$^-$), is $\frac{m}{n} = 1.2 \pm 0.5$.

## The release of arsenic into Bangladesh's drinking well water by the salt effect, and the expected impact of sea level rise

**Background.** The salt effect, diverse ion effect, or uncommon ion effect describes the increase in the solubility of an ionic solid, such as a mineral, when it is in a solution of an electrolyte that does not have any ions that are in common with the solid. Barium sulfate (BaSO$_4$) is not a concern in Bangladesh; however, it has been exhaustively studied in the laboratory, so it is used here to explain the salt effect. The solubility of solid barium sulfate (BaSO$_{4(s)}$) increases as the concentration of dissolved potassium nitrate (K$^+_{(aq)}$ + NO$_3^-_{(aq)}$) increases. That is, K$^+_{(aq)}$ and NO$_3^-_{(aq)}$ shifts the following equilibrium to the right; this increases the solubility of solid barium sulfate (BaSO$_{4(s)}$; Reactions 5 and 6) [27, 28]. In short, other ions dissolved in water will make an ionic solid more soluble.

$$\text{Without diverse ions}: \quad BaSO_{4(s)} \rightleftharpoons Ba^{2+}_{(aq)} + SO_{4(aq)}^{2-} \tag{5}$$

$$\text{With diverse ions}: \quad BaSO_{4(s)} + K^+_{(aq)} + NO_{3(aq)}^- \rightleftharpoons Ba^{2+}_{(aq)} + SO_{4(aq)}^{2-} + K^+_{(aq)} + NO_{3(aq)}^- \tag{6}$$

The salt effect is driven by the electromagnetic force, the attraction of oppositely charged particles to each other [29]. Therefore, it is essential to know if a chemical is positively charged, negatively charged, or electrically neutral. Positively charged ions are attracted to negatively charged and can take part in the salt effect. Electrically neutral chemicals cannot take part in the salt effect.

The salt effect has two processes. The first process of the salt effect is ion exchange, the displacement of adsorbed ions on solid surfaces by dissolved ions in water [21]. For example, aqueous potassium ions (K$^+_{(aq)}$) exchange with adsorbed barium ions (Ba$^{2+}_{(ex)}$) on the surface of solid barium sulfate (BaSO$_{4(s)}$; Reaction 6). Also, aqueous nitrate ions (NO$_3^-_{(aq)}$) exchange with adsorbed sulfate ions (SO$_4^{2-}_{(ex)}$) on the surface of solid barium sulfate (BaSO$_{4(s)}$ Reaction 6). This increases the solubility of barium sulfate (BaSO$_{4(s)}$). (The phase designation "ex" denotes "exchangeable").

The second process of the salt effect is ion pairing [30]. The diverse ions from a dissolved salt pair with the dissolved ions from an ionic solid. For example, aqueous potassium ions (K$^+_{(aq)}$) pair with oppositely charged aqueous sulfate ions (SO$_4^{2-}_{(aq)}$) in solution to make [KSO$_4]^-_{(aq)}$, remove product, and shift this equilibrium to the right (Reaction 6). Also, aqueous nitrate ions (NO$_3^-_{(aq)}$) pair with oppositely barium ions (Ba$^{2+}_{(aq)}$) in solution to make, [BaNO$_3]^+_{(aq)}$, remove additional product, and shift this equilibrium further to the right (Reaction 6).

Arsenic is also released from these naturally occurring minerals into drinking well water by the salt effect. The salt effect is driven by the electromagnetic force, the attraction of oppositely charged particles to each other [29]. Therefore, it is essential to know if a chemical is positively charged, negatively charged, or electrically neutral. Positively charged ions (cations) are

attracted to negatively charged ions (anions) and can take part in the salt effect. Electrically neutral chemicals cannot take part in the salt effect. Since arsenic in groundwater commonly exists as weak acids, it is also essential to know how their charges change with pH.

In saline environments, oxyanions of arsenic are released from the surfaces of these iron and manganese oxyhydroxides into groundwater by hydroxide ($OH^-_{(aq)}$), chloride ($Cl^-_{(aq)}$), bicarbonate ($HCO_3^-_{(aq)}$), phosphate ($H_{3-x}PO_4^{x-}_{(aq)}$), and other typical aqueous anions [8, 19, 20, 31–33]. Arsenic acid ($H_3As(V)O_4$; p$K_{a1}$ = 2.25; Fig 8) [23] is a stronger acid than arsenous acid ($H_3As(III)O_3$; p$K_{a1}$ = 9.23; Fig 9) [23]; therefore, from pH 2.25 to pH 9.23, arsenate ($H_{3-x}As(V)O_4^{x-}$) mostly has either a −1 or −2 net charge, and arsenite is mostly fully protonated and electrically neutral ($H_3As(III)O_3$) [23, 34]. Additional information regarding the dissociation of $H_3As(V)O_4$ and $H_3As(III)O_3$ as a function of pH is included in the S4–S6 Files.

The iron and manganese oxyhydroxide surfaces in Bangladesh's aquifer have a pH dependent charge [8]. At lower pH values, these surfaces have a positive net charge and arsenic is mostly adsorbed as a negatively charged arsenate ($H_{3-x}As(V)O_4^{x-}$) anion (Fig 8) [8, 23]. In contrast, the electrically neutral arsenite ($H_3As(III)O_3$) is rarely absorbed at lower pH values

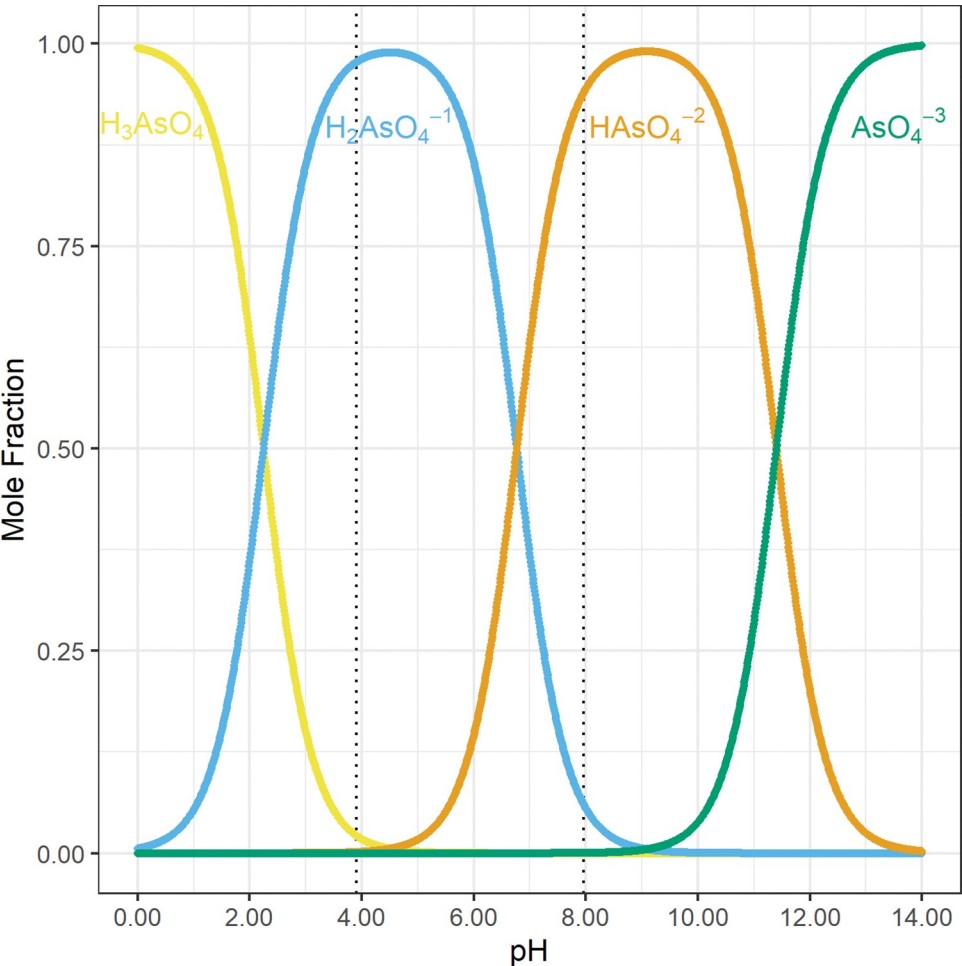

**Fig 8. The mole fraction of arsenic acid ($H_3As(V)O_4$) species as a function of pH.** These concentrations were calculated from published equilibrium constants S4 File. The dotted vertical lines at pH = 3.90 and pH = 7.96 demarcate the range of pH values observed in this study. This suggests that from the minimum observed pH at 3.90 to pH = p$K_{a2}$ at 6.77 most of the As(V) has a −1 charge, and from pH = p$K_{a2}$ at 6.77 to the maximum observed pH at 7.96 most of the As(V) has a −2 charge [23].

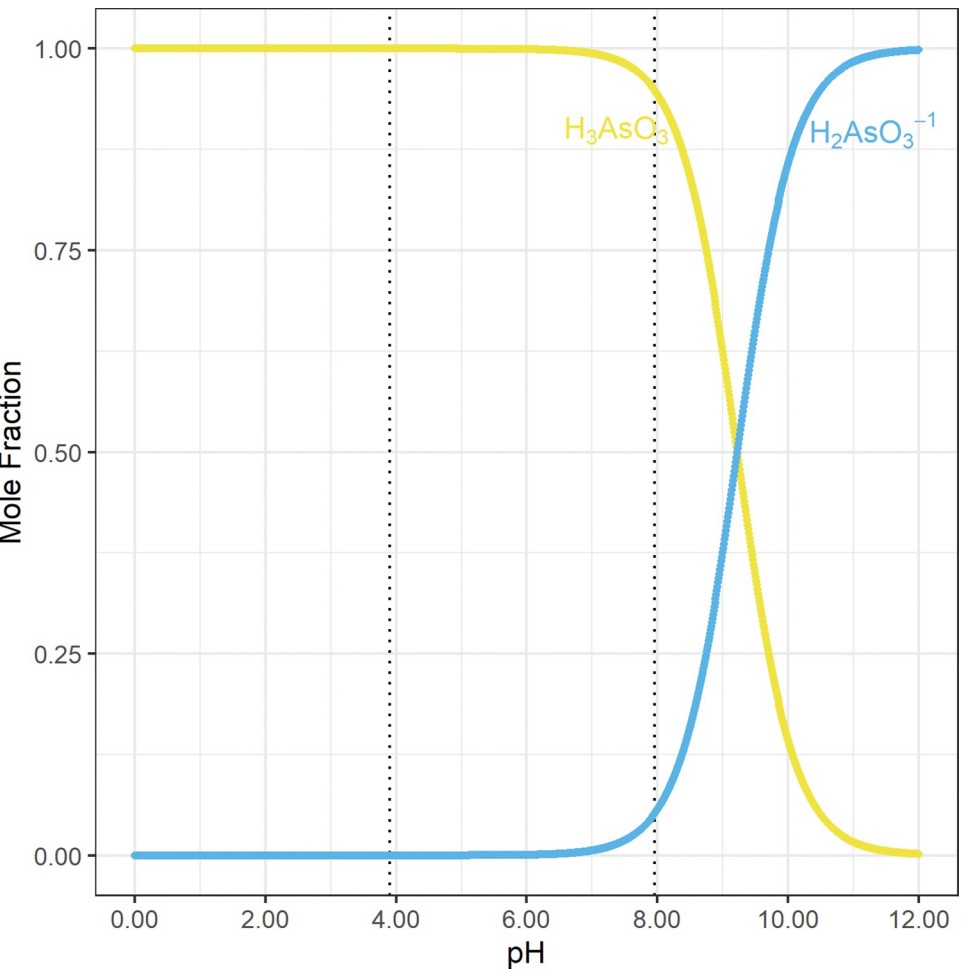

**Fig 9. The mole fraction of arsenous acid (H₃As(III)O₃) species as a function of pH.** These concentrations were calculated from published equilibrium constants S4 File. The dotted vertical lines at pH = 3.90 and pH = 7.96 demarcate the range of pH values observed in this study. This suggests that from the minimum observed pH at 3.90 to the maximum observed pH at 7.96 most of the As(III) is electrically neutral [23].

(Fig 9) [8, 23]. As the pH increases, the net positive surface charge of these minerals decreases and arsenic is released into groundwater, even in an oxidizing environment [8]. This release of adsorbed arsenic from the surface of minerals into groundwater is increased by hydroxide ($OH^-_{(aq)}$), chloride ($Cl^-_{(aq)}$), bicarbonate ($HCO_3^-_{(aq)}$), phosphate ($H_{3-x}PO_4^{x-}_{(aq)}$), and other typical aqueous anions [8, 19, 20, 31–33].

**Specific conductance and the release of arsenic.** As sea levels continue to rise and floods and cyclones in Bangladesh increase in area and duration, the underlying aquifer can become more saline. The following data strongly suggest that this chemical change can increase the release of arsenic into Bangladesh's drinking well water by the salt effect.

Specific conductance measures the ability of an aqueous solution to carry an electrical current; it depends on the concentrations and charges of the ions that are dissolved in water. Specific conductance measures the inverse of electrical resistance between two inert electrodes; its units are microsiemens per centimeter (μS/cm). Specific conductance is used to measure the salinity of water. In general, the greater the specific conductance the greater the salinity of water [16]. In general, the greater the salinity of water the greater the salt effect.

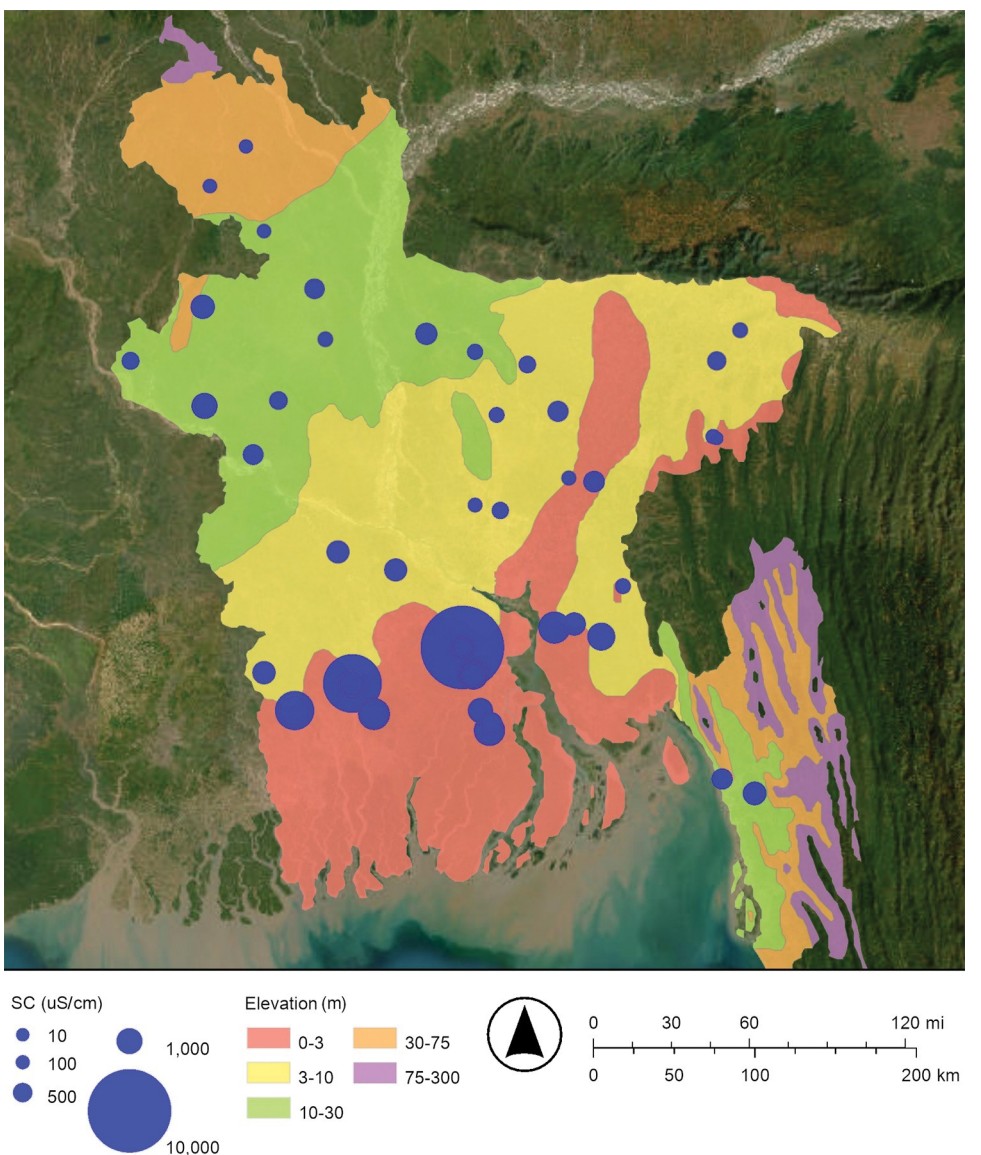

**Fig 10. Proportional symbol map of specific conductance (SC) in microsiemens per centimeter (μS/cm) of Bangladesh's drinking well water.** The map was created with ArcGIS® Online using a public domain basemap published by the United States Geological Survey [15].

Maps of the concentration of arsenic and the specific conductance of Bangladesh's drinking well water are shown in Figs 2 and 10, respectively. These maps suggest that when the specific conductance increases, the concentration of arsenic increases. The specific conductance of drinking well water in this study ranged from 40 μS/cm to 5,360 μS/cm (Fig 10; Table 1). For comparison, the conductance of seawater at the ocean's surface is about 50,000 μS/cm [35]. The map of the specific conductance of Bangladesh's drinking well water has higher specific conductance in the south near the Bay of Bengal and lower specific conductance in the north near the Himalayan mountains; this is consistent with floods and storm surges causing relatively higher specific conductance saltwater to intrude from the south to the north (Fig 10).

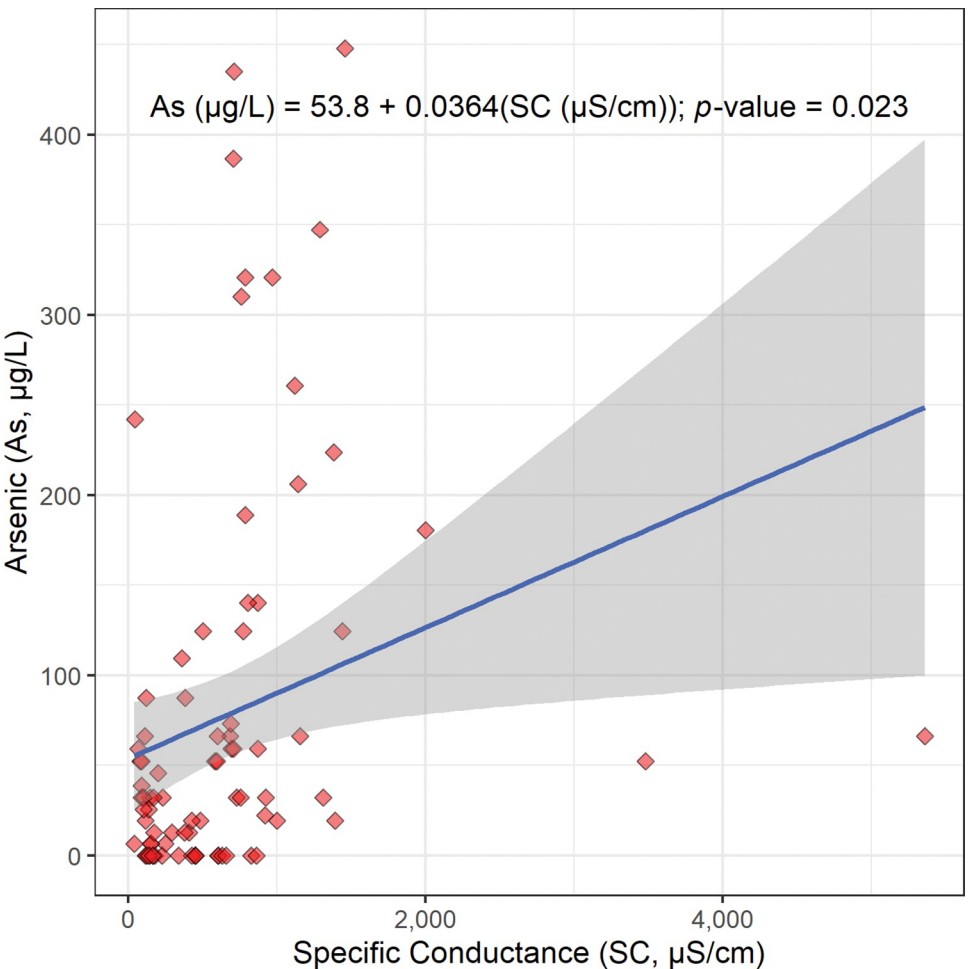

**Fig 11. The scatterplot, linear regression equation, 95% confidence band, and *p*-value for the concentration of arsenic (As) in micrograms per liter (µg/L) versus specific conductance (SC) in microsiemens per centimeter (µS/cm).** This regression is statistically significant at a Benjamini and Hochberg's False Discovery Rate corrected 95% confidence level (Table 2) [18].

The scatterplot, linear regression equation, and *p*-value for the concentration of arsenic versus specific conductance are shown in Fig 11. This linear regression gives a statistically significant positive slope (0.0364 µg cm/L µS; Fig 11; Table 2). This suggests that arsenic is released from sediments into Bangladesh's drinking well water by the salt effect. More specifically, this positive relationship suggests that any process that increases salinity, such as annual flooding, is expected to increase the release of arsenic oxyanions from sediments into Bangladesh's drinking well water by the salt effect [36].

Ions that are dissolved in water can only interact with solid surfaces and with other ions that are dissolved in water. As a result, there are only two components of the salt effect: the displacement of ions from solid surfaces by ion exchange by ions in water, and the pairing of oppositely charged ions in water. The first component of the salt effect is ion exchange, the displacement of adsorbed ions on solid surfaces by dissolved ions in water [21]. The anion exchange component of the salt effect is shown in simplified Reaction 7. This reaction is simplified in that it uses only monoprotonated arsenate ($HAs(V)O_4^{2-}$) as a model oxyanion of arsenic. That is, for the sake of simplicity, only one combination of oxidation state and degree

**Table 3. The mole fraction of arsenic acid ($H_3As(V)O_4$) species at pH 3.90 and pH 7.96, the minimum and maximum pH values observed in this study ([Fig 8](); [Table 1]()) [23].**

| pH | $H_3As(V)O_{4(aq)}$ | $H_2As(V)O_4^-{}_{(aq)}$ | $HAs(V)O_4^{2-}{}_{(aq)}$ | $As(V)O_4^{3-}{}_{(aq)}$ |
|---|---|---|---|---|
| 3.90 | 0.022 | 0.98 | 0.0013 | $4.1 \times 10^{-11}$ |
| 7.96 | $1.2 \times 10^{-7}$ | 0.061 | 0.94 | 0.00034 |

of protonation is shown.

$$[\text{Solid Surface}]^{2+}\text{HAs(V)O}_{4(ex)}^{2-} + 2\text{Cl}^-_{(aq)} \rightleftarrows [\text{Solid Surface}]^{2+}(\text{Cl}^-)_{2(ex)} + \text{HAs(V)O}_{4(aq)}^{2-} \qquad (7)$$

More specifically, oxyanions of arsenate ($H_{3-x}As(V)O_4^{x-}$) and possibly arsenite ($H_{3-x}As(III)O_3^{x-}$) are likely displaced from positively charged solid surfaces in Bangladesh's aquifer by anion exchange with aqueous chloride ($Cl^-_{(aq)}$), hydroxide ($OH^-_{(aq)}$), bicarbonate ($HCO_3^-{}_{(aq)}$), phosphate ($H_{3-x}PO_4^{x-}{}_{(aq)}$), and other anions in the seawater that are deposited by the annual floods [8, 20, 25].

The salt effect is driven by the electromagnetic force, the attraction of oppositely charged particles to each other [29]. For example, the positively charged solid surface ($[\text{Solid Surface}]^{2+}$) and the negatively changed monohydrogen arsenate ion ($HAs(V)O_4^{2-}$) in Reaction 7 are attracted to each other. The larger the difference in positive and negative charge, the stronger the attraction at a given distance [29]. That is, the attraction between monohydrogen arsenate ion ($HAs(V)O_4^{2-}$) and the solid surface ($[\text{Solid Surface}]^{2+}$) in Reaction 7 is approximately 2 times greater than the attraction between chloride ion ($Cl^-$) and the solid surface ($[\text{Solid Surface}]^{2+}$) in Reaction 7.

The charges of the polyvalent anions from arsenic acid ($H_3As(V)O_4$; [Fig 8]()) [23] and arsenous acid ($H_3As(III)O_3$; [Fig 9]()) [23] change with pH. Therefore, the charges of these polyvalent anions at different pH values must be identified to assess the release of arsenic by anion exchange.

Arsenic acid ($H_3As(V)O_4$; $pK_{a1} = 2.25$; [Fig 8]()) [23] is a stronger acid than arsenous acid ($H_3As(III)O_3$; $pK_{a1} = 9.23$; [Fig 9]()) [23]; therefore, at range of pH values observed in this study, pH 3.90 to 7.96, arsenate ($H_{3-x}As(V)O_4^{x-}$) mostly has either a −1 or −2 net charge ([Fig 8](); [Table 3]()), and arsenite is mostly fully protonated and electrically neutral ($H_3As(III)O_3$; [Fig 9](); [Table 4]()) [23]. More specifically, 98% of arsenate ($H_{3-x}As(V)O_4^{x-}$) has a −1 net charge at pH 3.90, and 94% of arsenate ($H_{3-x}As(V)O_4^{x-}$) has a −2 net charge at pH 7.96 ([Table 3]()). In contrast, only 0.00048% of arsenite ($H_{3-x}As(III)O_3^{x-}$) has a −1 net charge at pH 3.90, and only 5.2% of arsenite ($H_{3-x}As(III)O_3^{x-}$) has a −1 net charge at pH 7.96 ([Table 4]()).

Electrically neutral chemicals cannot take part in the salt effect. Arsenate ($H_{3-x}As(V)O_4^{x-}$) is more likely to be negatively charged than arsenite ($H_{3-x}As(III)O_3^{x-}$); therefore, As(V) is more likely to be involved in anion exchange than As(III). However, the adsorption of electrically neutral As(III) onto solid surfaces by hydrogen bonding and other dipole forces is highly likely [29]; therefore, the displacement of neutral As(III) from solid surfaces and into Bangladesh's drinking well water by dissolved ions is also possible. Technically, this possible

**Table 4. The mole fraction of arsenous acid ($H_3As(III)O_3$) species at pH 3.90 and pH 7.96, the minimum and maximum pH values observed in this study ([Fig 9](); [Table 1]()) [23].**

| pH | $H_3As(III)O_3(aq)$ | $H_2As(III)O_3^-(aq)$ |
|---|---|---|
| 3.90 | 1.0 | $4.8 \times 10^{-6}$ |
| 7.96 | 0.95 | 0.052 |

displacement of neutral As(III) would not be anion exchange, but it would still release arsenic into Bangladesh's drinking well water.

The second component of the salt effect is ion pairing, the electrostatic attraction of oppositely charged ions in water [30]. Ionic solids, such as the minerals in Bangladesh's aquifer, are in equilibrium with their dissolved ions in water. The pairing of these dissolved ions with oppositely charged ions from saltwater shifts this equilibrium; this shift favors the release of arsenic into Bangladesh's drinking well water. This ion pairing component of the salt effect is shown in simplified sequential Reactions 8 and 9. For the sake of simplicity, only one combination of oxidation state and degree of protonation is shown.

$$Fe(III)As(V)O_{4(s)} \rightleftharpoons Fe^{3+}_{(aq)} + As(V)O^{3-}_{4(aq)} \tag{8}$$

$$Fe^{3+}_{(aq)} + As(V)O^{3-}_{4(aq)} + Na^{+}_{(aq)} + Cl^{-}_{(aq)} \rightleftharpoons Fe(III)Cl^{2+}_{(aq)} + NaAs(V)O^{2-}_{4(aq)} \tag{9}$$

More specifically, oxyanions of arsenate ($H_{3-x}As(V)O_4^{x-}$) and possibly arsenite ($H_{3-x}As(III)O_3^{x-}$) are likely released from solid sediments in Bangladesh's aquifer by ion pairing. That is, the equilibrium between a solid ionic reactant and its dissolved ions is shifted toward products. In this example, the shift in equilibrium is caused by the removal of dissolved ions by ion pairing with aqueous sodium ($Na^{+}_{(aq)}$), chloride ($Cl^{-}_{(aq)}$), and other ions in the seawater deposited during annual flooding.

Ion pairing not only shifts the equilibrium to the right by removing products; it also shifts the equilibrium to the right by inhibiting the reformation of ionic solid. For example, the loss in net positive charge from the pairing of $Fe^{3+}_{(aq)}$ and $Cl^{-}_{(aq)}$ to make $[Fe(III)Cl]^{2+}_{(aq)}$, and the loss in net negative charge from the pairing of $Na^{+}_{(aq)}$ and $As(V)O_4^{3-}_{(aq)}$ to make $[NaAs(V)O_4]^{2-}_{(aq)}$ decreases the electrostatic attraction that is needed to precipitate $Fe(III)As(V)O_{4(s)}$. This favors the release of arsenic from the ionic solid into Bangladesh's drinking well water.

In conclusion, the area and duration of saltwater intrusion into Bangladesh's aquifer is expected to increase as sea levels rise from climate change. The increase in salinity is expected to increase the release of arsenic oxyanions from sediments into Bangladesh's drinking well water by the salt effect.

**pH and the release of arsenic.** Specific conductance is a measurement of the concentration of all ions in water; in contrast, pH is the measurement of the concentration of a single ion, aqueous hydrogen ion ($H^{+}_{(aq)}$). In addition, this measurement of pH allows the calculation of the concentration of aqueous hydroxide ion ($OH^{-}_{(aq)}$) [24].

Maps of the concentration of arsenic and the pH of Bangladesh's drinking well water are shown in Figs 2 and 12, respectively. These maps suggest that when the pH increases, the concentration of arsenic increases. The pH of drinking well water in this study ranged from pH 3.90 to pH 7.96 (Fig 12; Table 1). The average pH of oceanwater is pH 8.1 [37]. The map of the pH of Bangladesh's drinking well water has higher pH values in the south near the Bay of Bengal and lower pH values in the north near the Himalayan mountains; this is consistent with floods and storm surges causing higher pH saltwater to intrude from the south to the north (Fig 12).

The scatterplot, linear regression equation, and *p*-value for the concentration of arsenic versus pH are shown in Fig 13. This linear regression gives a statistically significant positive slope (36.3 μg/L pH unit; Fig 13; Table 2). This positive relationship between arsenic concentration and pH suggests that arsenic is released from sediments into Bangladesh's drinking well water as the pH increases.

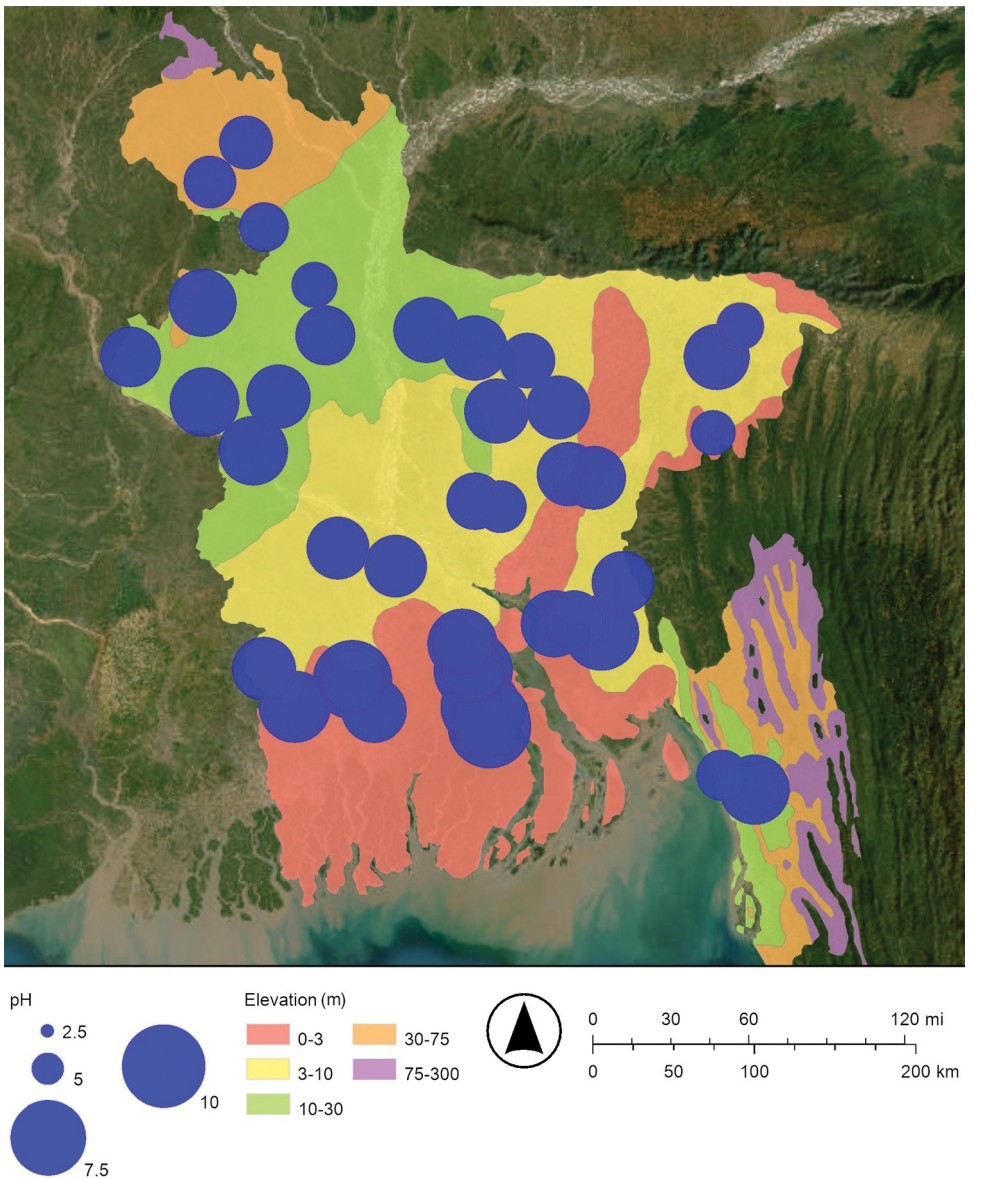

**Fig 12. Proportional symbol map of the pH of Bangladesh's drinking well water.** The map was created with ArcGIS® Online using a public domain basemap published by the United States Geological Survey [15].

This relationship can be explained the salt effect. Oxyanions of arsenate ($H_{3-x}As(V)O_4^{x-}$) and possibly arsenite ($H_{3-x}As(III)O_3^{x-}$) are most likely displaced from positively charged solid surfaces in Bangladesh's aquifer by anion exchange with aqueous hydroxide ion ($OH^-_{(aq)}$) as the pH increases. This anion exchange component of the salt effect is shown in simplified Reaction 10. For the sake of simplicity, only one combination of oxidation state and degree of protonation is shown.

$$[\text{Solid Surface}]^{2+}HAs(V)O_{4(ex)}^{2-} + 2OH^-_{(aq)} \rightleftarrows$$

$$[\text{Solid Surface}]^{2+}(OH^-)_{2(ex)} + HAs(V)O_{4(aq)}^{2-} \tag{10}$$

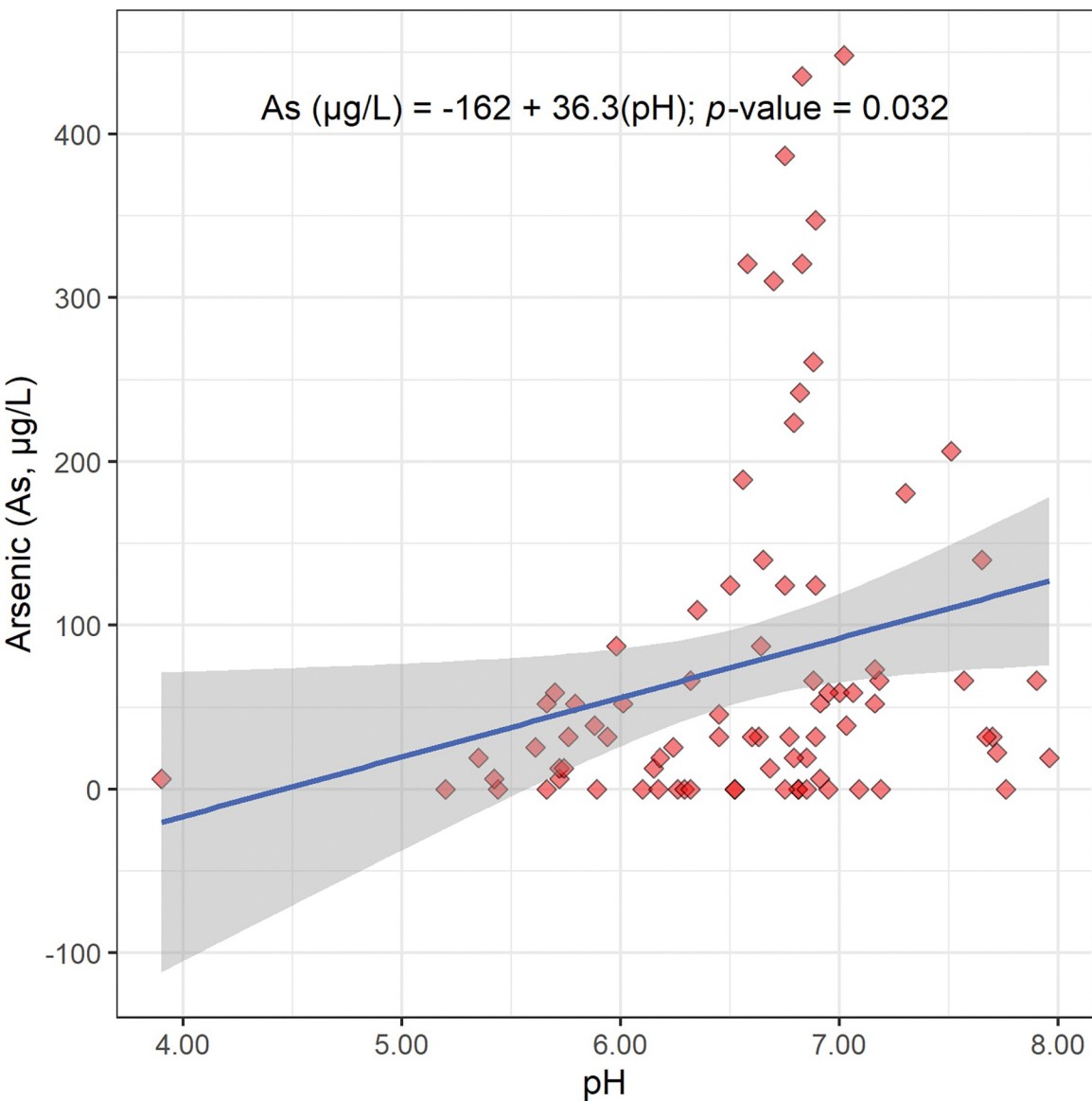

**Fig 13. The scatterplot, linear regression equation, 95% confidence band, and _p_-value for the concentration of arsenic (As) in micrograms per liter (µg/L) versus pH.** This regression is statistically significant at a Benjamini and Hochberg's False Discovery Rate corrected 95% confidence level (Table 2) [18].

The pairing of aqueous ions from the dissolution of an ionic solid that contains arsenic would also favor the release of arsenic into Bangladesh's drinking well water as the pH increases. This ion pairing component of the salt effect is shown in simplified sequential Reactions 8 and 11. In this example, the negative charge from aqueous hydroxide ion ($OH^-_{(aq)}$) is balanced by a positive charge of aqueous sodium ion ($Na^+_{(aq)}$) to maintain the electroneutrality of the solution (Reaction 11). For the sake of simplicity, only one combination of oxidation state and degree of protonation is shown.

$$Fe(III)As(V)O_{4(s)} \rightleftharpoons Fe^{3+}_{(aq)} + As(V)O^{3-}_{4(aq)} \tag{8}$$

$$Fe_{(aq)}^{3+} + As(V)O_{4(aq)}^{3-} + Na_{(aq)}^{+} + OH_{(aq)}^{-} \rightleftharpoons Fe(III)OH_{(aq)}^{2+} + NaAs(V)O_{4(aq)}^{2-} \tag{11}$$

In conclusion, the area and duration of saltwater intrusion into Bangladesh's aquifer is expected to increase as sea levels rise from climate change. This increased saltwater intrusion is expected to increase the pH of the underlying aquifer and increase the release of arsenic oxyanions from sediments into Bangladesh's drinking well water by the salt effect.

**The release of arsenic into Bangladesh's drinking well water with increasing temperature.** The change in Gibbs free energy ($\Delta G$) determines if a chemical reaction is thermodynamically favorable or not. If $\Delta G$ is negative, then the reaction is thermodynamically favorable and is expected to spontaneously go from reactants to products. If $\Delta G$ is positive, then the reaction is not thermodynamically favorable and cannot spontaneously go from reactants to products [24].

The $\Delta G$ of a reaction equals the change in enthalpy ($\Delta H$) minus the temperature (T) in kelvins (K) times the change in entropy ($\Delta S$). That is, $\Delta G = \Delta H - T\Delta S$. The temperature in kelvin is always positive. Dissolving of an ionic solid releases its ions into solution; this increases randomness and makes $\Delta S$ positive [24].

Therefore, for a dissolving ionic solid, a positive T times a positive $\Delta S$ makes $- T\Delta S$ negative. This negative $- T\Delta S$ contributes to a favorable $\Delta G$. As the temperature increases, this favorable $-T\Delta S$ contributes to an increasingly favorable $\Delta G$.

In other words, the solubility of a mineral in water should increase with temperature. If this mineral contains arsenic, then the release of arsenic into drinking well water should also increase as the temperature increases.

Maps of the concentration of arsenic and the temperature of Bangladesh's drinking well water are shown in Figs 2 and 14, respectively. These maps suggest that there is not a strong relationship between the narrow range of groundwater temperatures in this study (24.3˚C to 32.7˚C) and the concentration of arsenic (Table 1).

The scatterplot, linear regression equation, and $p$-value for the concentration of arsenic versus temperature are shown in Fig 15. As expected, this linear regression does have a positive slope; however, this slope is not statistically significant (9.31 µg/L ˚C; Fig 15; Table 2). A possible reason why this expected relationship was not statistically significant is that the range of groundwater temperatures in this study (24.3˚C to 32.7˚C) was too narrow to give a statistically significant increase in the concentration of arsenic with temperature (Table 1).

## Conclusions

As sea levels continue to rise due to climate change and the annual floods and cyclones in Bangladesh increase in area and duration, chemical processes will increase the release of arsenic into Bangladesh's drinking well water. This will increase the incidence of chronic arsenic poisoning. These chemical processes are reduction and the salt effect.

Tens of millions of people in Bangladesh are drinking well water with arsenic concentrations that are greater than the 10 µg/L World Health Organization guideline. This large exposure to arsenic in Bangladesh is increasing the rates of death and disease from skin, bladder, liver, and lung cancers and vascular disease; this is a public health crisis [11–13].

The source of the arsenic is the alluvial sediments that have been deposited by the Ganges, Brahmaputra, and Meghna rivers (Fig 1) [8]. These sediments release arsenic from a mixture of iron oxyhydroxide, manganese oxyhydroxide, and possibly other minerals into Bangladesh's drinking well water [8, 19, 20].

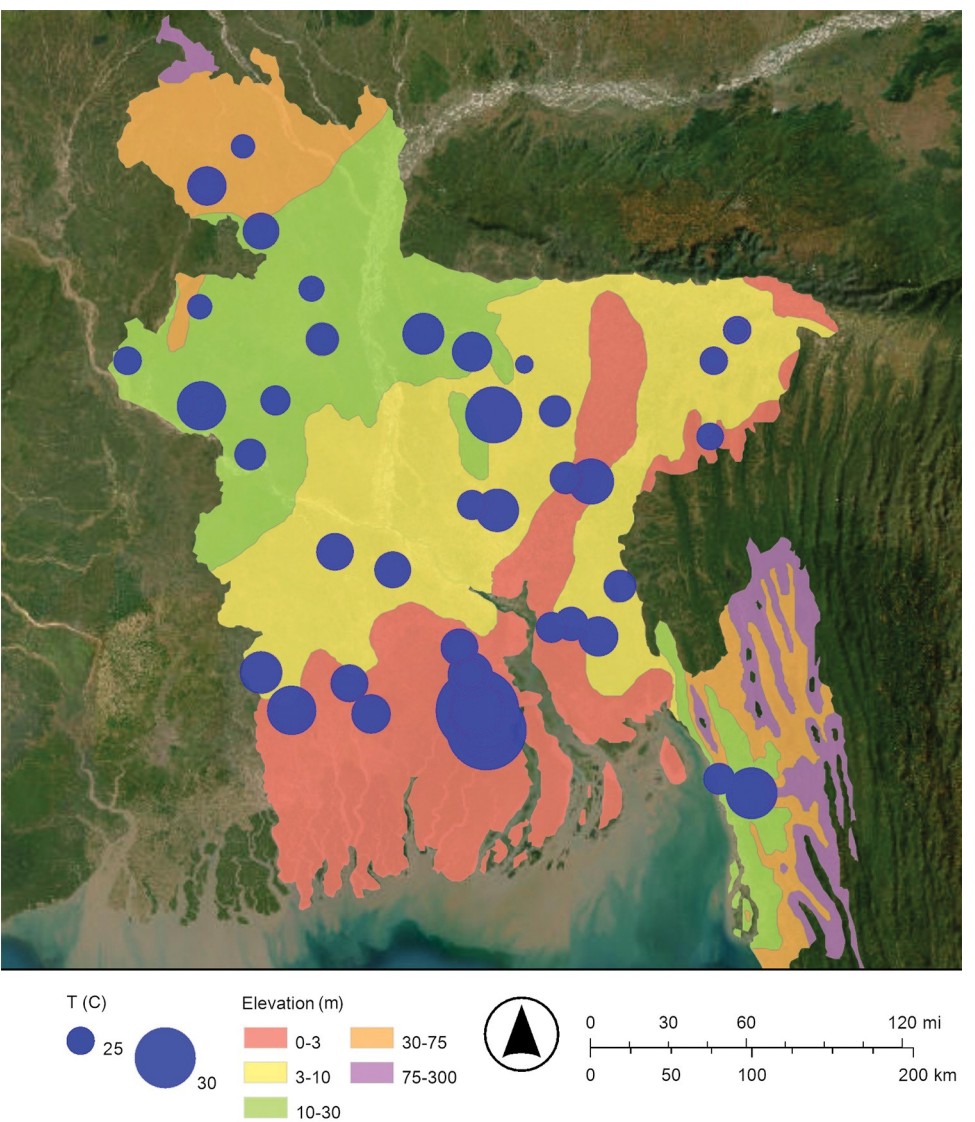

**Fig 14. Proportional symbol map of the temperature (T) in Celsius (˚C) of Bangladesh's drinking well water.** The map was created with ArcGIS® Online using a public domain basemap published by the United States Geological Survey [15].

In this study, the regression of arsenic concentration versus dissolved oxygen concentration gives a statistically significant negative slope (−14.1 μg/mg; Figs 2–4; Table 2). Also, the regression of arsenic concentration versus oxidation-reduction potential gives a statistically significant negative slope (−0.454 μg/L mV, Figs 2, 5 and 6; Table 2). These results are consistent with arsenic being released from sediments into Bangladesh's drinking well water by reduction.

The regression of arsenic concentration versus specific conductance gives a statistically significant positive slope (0.0364 μg cm/L μS; Figs 2, 10 and 11; Table 2). Also, the regression of arsenic concentration versus pH gives a statistically significant positive slope (36.3 μg/L pH unit; Fig 2, 12 and 13; Table 2). These results are consistent with arsenic being released from sediments into Bangladesh's drinking well water by the salt effect.

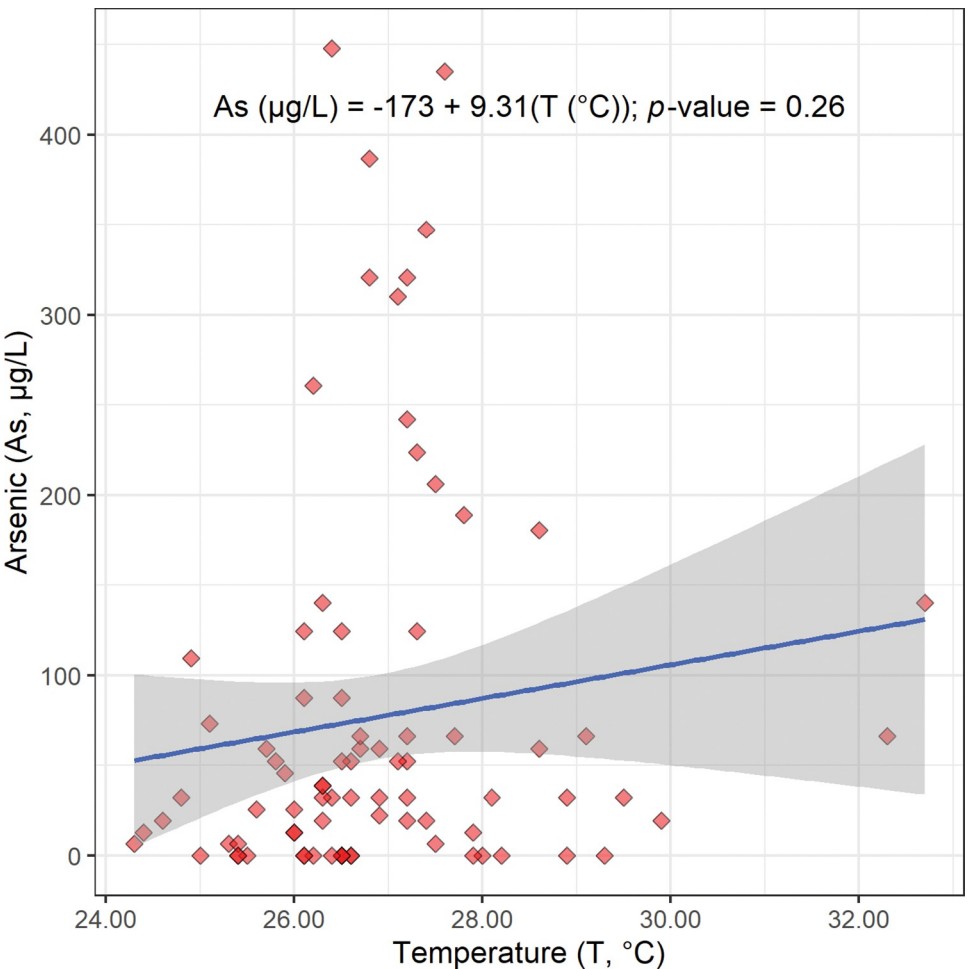

**Fig 15. The scatterplot, linear regression equation, 95% confidence band, and *p*-value for the concentration of arsenic (As) in micrograms per liter (µg/L) versus temperature (T) in Celsius (˚C).** This regression is not statistically significant at a Benjamini and Hochberg's False Discovery Rate corrected 95% confidence level (Table 2) [18].

In a typical monsoon season about 21% of Bangladesh's land is flooded with a mixture of freshwater from its rivers and saltwater from the Bay of Bengal [4]. As climate change progresses and sea levels continue to rise, the area and duration of these annual floods can increase. This increase in flooding is expected to decrease the oxidation-reduction potential and increase the salinity of Bangladesh's aquifer. These changes in aquifer chemistry are expected to increase the release of arsenic into Bangladesh's drinking well water by reduction and by the salt effect. Finally, this increased exposure to arsenic is expected to increase the rates of death and disease from chronic arsenic poisoning.

## Supporting information

**S1 Checklist. Inclusivity in global research questionnaire.**
(DOCX)

**S1 File. Raw data, calculations, and independent statistical analyses.**
(XLSX)

**S2 File. The R code that was used to analyze the arsenic concentration, dissolved oxygen concentration, oxidation-reduction potential, specific conductance, pH, and temperature data in S1 File.**
(R)

**S3 File. The derivation of a pH-dependent Nernst equation.**
(DOCX)

**S4 File. Acid dissociation data for $H_3As(V)O_4$ and $H_3As(III)O_3$.**
(XLSX)

**S5 File. The R code that was used to graph the acid dissociation data for $H_3As(V)O_4$ in S4 File.**
(R)

**S6 File. The R code that was used to graph the acid dissociation data for $H_3As(III)O_3$ in S4 File.**
(R)

## Acknowledgments

We thank Marwan Said Abualrub, Ph.D., and Amrutaa Vibho for reviewing our mathematical application of the Nernst equation.

## Author Contributions

**Conceptualization:** Seth H. Frisbie.

**Formal analysis:** Seth H. Frisbie, Erika J. Mitchell.

**Investigation:** Seth H. Frisbie, Erika J. Mitchell, Azizur R. Molla.

**Methodology:** Seth H. Frisbie, Erika J. Mitchell.

**Project administration:** Seth H. Frisbie.

**Resources:** Seth H. Frisbie.

**Supervision:** Seth H. Frisbie.

**Validation:** Seth H. Frisbie, Erika J. Mitchell, Azizur R. Molla.

**Visualization:** Seth H. Frisbie.

**Writing – original draft:** Seth H. Frisbie.

**Writing – review & editing:** Seth H. Frisbie, Erika J. Mitchell, Azizur R. Molla.

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
