## [Decision Letter · Decision Letter 0]

10 Oct 2023

PONE-D-23-21817Sea level rise from climate change is expected to increase the release of arsenic into Bangladesh’s drinking well water by reduction and by the salt effectPLOS ONE

Dear Dr. Frisbie,

Thank you for submitting your manuscript to PLOS ONE. After careful consideration, we feel that it has merit but does not fully meet PLOS ONE’s publication criteria as it currently stands. Therefore, we invite you to submit a revised version of the manuscript that addresses the points raised during the review process.

We look forward to receiving your revised manuscript.

Kind regards,

Venkatramanan Senapathi, Ph.D.

Academic Editor

PLOS ONE

Journal Requirements:

2. Please include the following request in the decision letter, and ping me with follow up. “Please include a complete copy of PLOS’ questionnaire on inclusivity in global research in your revised manuscript. Our policy for research in this area aims to improve transparency in the reporting of research performed outside of researchers’ own country or community. The policy applies to researchers who have travelled to a different country to conduct research, research with Indigenous populations or their lands, and research on cultural artefacts. The questionnaire can also be requested at the journal’s discretion for any other submissions, even if these conditions are not met.  Please find more information on the policy and a link to download a blank copy of the questionnaire here: https://journals.plos.org/plosone/s/best-practices-in-research-reporting. Please upload a completed version of your questionnaire as Supporting Information when you resubmit your manuscript.

"The field work in Bangladesh was funded by the United States Agency of International Development (contract number US AID RE III 388-0070). This fieldwork began in July of 1997 and ended in August of 1997. All other costs have been paid from the personal savings of the authors. We are grateful to Marwan Said Abualrub, Ph.D. for reviewing our mathematical application of the Nernst equation."

"The fieldwork in Bangladesh was funded by the United States Agency of International Development (USAID; contract number US AID RE III 388-0070; https://www.usaid.gov/). This fieldwork began in July of 1997 and ended in August of 1997. USAID paid Seth H. Frisbie's (SHF) salary during these two months in 1997. All other costs have been paid from the personal savings of the authors. The funders had no role in study design, data collection and analysis, decision to publish, or preparation of the manuscript."

5. We note that Figures 1, 5, 6, 8, 11, 13, 15 in your submission contain [map/satellite] images which may be copyrighted. All PLOS content is published under the Creative Commons Attribution License (CC BY 4.0), which means that the manuscript, images, and Supporting Information files will be freely available online, and any third party is permitted to access, download, copy, distribute, and use these materials in any way, even commercially, with proper attribution. For these reasons, we cannot publish previously copyrighted maps or satellite images created using proprietary data, such as Google software (Google Maps, Street View, and Earth). For more information, see our copyright guidelines: http://journals.plos.org/plosone/s/licenses-and-copyright.

1.) You may seek permission from the original copyright holder of Figures 1, 5, 6, 8, 11, 13, 15 to publish the content specifically under the CC BY 4.0 license.  

2.) If you are unable to obtain permission from the original copyright holder to publish these figures under the CC BY 4.0 license or if the copyright holder’s requirements are incompatible with the CC BY 4.0 license, please either i) remove the figure or ii) supply a replacement figure that complies with the CC BY 4.0 license. Please check copyright information on all replacement figures and update the figure caption with source information. If applicable, please specify in the figure caption text when a figure is similar but not identical to the original image and is therefore for illustrative purposes only.

Reviewers' comments:

Reviewer's Responses to Questions

**Comments to the Author**

1. Is the manuscript technically sound, and do the data support the conclusions?

Reviewer #1: Yes

Reviewer #2: Yes

2. Has the statistical analysis been performed appropriately and rigorously? 

Reviewer #1: Yes

Reviewer #2: I Don't Know

3. Have the authors made all data underlying the findings in their manuscript fully available?

Reviewer #1: Yes

Reviewer #2: No

4. Is the manuscript presented in an intelligible fashion and written in standard English?

Reviewer #1: Yes

Reviewer #2: Yes

5. Review Comments to the Author

Reviewer #1: The paper by Frisbie et al. is an interesting paper on the connection between sea level rise and water quality beyond just salinization of coastal aquifers. It brings to the reader a known problem in Bangladesh but under a different light.

The paper requires a complete restructuring and rewriting of the sections. Moreover, figures should be combined given the similarities. It is also too long and the statistical analysis of the data cannot be related to the detailed geochemistry reported.

The manuscript needs to be re-written in a simpler structure including sections such as:

Introduction, materials and methods reporting the sites and the statistical analysis and the analytical methods, a section on the geochemistry where the relationships between the salinity and the field data are highlighted to support the hypotheses, results and discussion with integrated figures (currently there are too many figures), and a conclusion which does not summarizes the paper but it really highlights the outcomes.

Below are few detailed comments.

Detailed comments

The introduction should be a single section starting from the overall problem of salinization of coastal aquifers and pollution or directly arsenic mobilization. Currently, it is too detailed regarding the problem of Bangladesh and it needs to be revised leaving detailed aspects of the site towards the end of the introduction.

Lines 79 and 82: the number should be written in letters. This is repeated throughout the paper. Please correct.

Figure 1 should be moved in material and methods and not in the introduction. The map should also report more details about the region to help the reader to identify locations such as coast, cities, and boundaries.

Figure 2 can be broad to the SI document and just described with words.

Line 219: I would instead write: ”Our research aims at identifying the relationship between the chemical changes of the coastal aquifer due to sea level rise along the coast of Bangladesh and the increase release of arsenic into drinking water wells”.

Line 269: Why was the “Benjamini and Hochberg’s False Discovery Rate” method chosen and not another one? A short description of the method and a justification of the choice should be included.

Section Mapping should be integrated with the description of the site.

Line 330 and 374 should be a section named geochemistry where the fundamental of the chemistry of arsenic are explained to support all the arguments raised by the measurements. This should be combined with section titled: The Nernst equation and the release of arsenic.

Line 897: I would bring up the relationship between conductivity and depth that are statistically significant and then that with oxygen.

The conclusion section should just briefly state the problem and summarize the major outcomes of the paper. It is instead a combination of abstract and manuscript summary.

Reviewer #2: The manuscript is about the "Sea level rise from climate change is expected to increase the release of arsenic into Bangladesh’s drinking well water by reduction and by the salt effect"

Some comments are suggested as below:

1. What was the sampling volume?

2. It seems that according to the measured concentration, the graphite furnace atomic absorption spectrophotometry (GFAA) is much more suitable.

3. rom any method of sampling, measurement has been done, it must be mentioned the source.

4. Please send me the R codes written for this article and its results in text file so that I can analyze it.

5. Summarize the relationship of different cases with arsenic auditors and compare with other articles.

6. How did you reach the arsenic ions by changing the pH? More explanation is needed.

7. The effects of arsenic in places where their levels are very high should be determined. Do you have any data in this field?

8. Considering that the amount of arsenic in Bangladesh is high and even reaches more than 40 times the standard in some places, describe its effects in Bangladesh using other studies that exist.

6. PLOS authors have the option to publish the peer review history of their article (what does this mean?). If published, this will include your full peer review and any attached files.

Reviewer #1: No

Reviewer #2: No

---

## [Author Response · Author response to Decision Letter 0]

23 Oct 2023

RESPONSE TO REVIEWERS

JOURNAL REQUIREMENT 1: Please ensure that your manuscript meets PLOS ONE’s style requirements, including those for file naming. The PLOS ONE style templates can be found at https://journals.plos.org/plosone/s/file?id=wjVg/PLOSOne_formatting_sample_main_body.pdf and https://journals.plos.org/plosone/s/file?id=ba62/PLOSOne_formatting_sample_title_authors_affiliations.pdf

RESPONSE TO JOURNAL REQUIREMENT 1: Done.

JOURNAL REQUIREMENT 2: Please include a complete copy of PLOS’ questionnaire on inclusivity in global research in your revised manuscript. Our policy for research in this area aims to improve transparency in the reporting of research performed outside of researchers’ own country or community. The policy applies to researchers who have travelled to a different country to conduct research, research with Indigenous populations or their lands, and research on cultural artefacts. The questionnaire can also be requested at the journal’s discretion for any other submissions, even if these conditions are not met. Please find more information on the policy and a link to download a blank copy of the questionnaire here: https://journals.plos.org/plosone/s/best-practices-in-research-reporting. Please upload a completed version of your questionnaire as Supporting Information when you resubmit your manuscript.

RESPONSE TO JOURNAL REQUIREMENT 2: Done. The completed Inclusivity in global research questionnaire was uploaded as Supporting Information when our manuscript was resubmitted. In addition, as requested, a subsection called “Inclusivity in global research” was added to the “Materials and methods” section as follows.

Inclusivity in global research

Additional information regarding the ethical, cultural, and scientific considerations specific to inclusivity in global research is included in the Supporting Information (S1 Checklist). This work was done at the request of the Government of Bangladesh by the United States Agency of International Development (USAID). More specifically, it was completed under the direction of the Bangladesh Rural Electrification Board. This study did not use animal or human subjects; therefore, the Government of Bangladesh did not require permits or approval from an ethics board.

Supporting Information

S1 Checklist. Inclusivity in global research questionnaire. (DOCX)

JOURNAL REQUIREMENT 3: Please note that PLOS ONE has specific guidelines on code sharing for submissions in which author-generated code underpins the findings in the manuscript. In these cases, all author-generated code must be made available without restrictions upon publication of the work. Please review our guidelines at https://journals.plos.org/plosone/s/materials-and-software-sharing#loc-sharing-code and ensure that your code is shared in a way that follows best practice and facilitates reproducibility and reuse.

RESPONSE TO JOURNAL REQUIREMENT 3: Done. Our R codes have been added to the Supporting Information section of the manuscript (please see S3 File.R, S6 File.R, and S7 File.R). In addition, citations to these R codes have been added to the body of the manuscript. The R codes have comments throughout to assist the readers in understanding our coding.

The revised Supporting Information section follows.

S1 Checklist. Inclusivity in global research questionnaire. (DOCX)

S2 File. Raw data, calculations, and independent statistical analyses. (XLSX)

S3 File. The R code that was used to analyze the arsenic concentration, dissolved oxygen concentration, oxidation-reduction potential, specific conductance, pH, and temperature data in S2 File.xlsx. (R)

S4 File. The derivation of a pH-dependent Nernst equation. (DOCX)

S5 File. Acid dissociation data for H3As(V)O4 and H3As(III)O3. (XLSX)

S6 File. The R code that was used to graph the acid dissociation data for H3As(V)O4 in S5 File.xlsx. (R)

S7 File. The R code that was used to graph the acid dissociation data for H3As(III)O3 in S5 File.xlsx. (R)

S4 File was included in response to a typographical error in the original manuscript; the 2.303 factor to convert from the thermodynamically relevant natural logarithm (ln) to the more convenient base 10 logarithm (log10) in the Nernst equation was accidentally omitted from Reaction 4. This typographical error is fixed in the current manuscript. All calculations in the original and current manuscripts correctly used this 2.303 conversion factor.

JOURNAL REQUIREMENT 4: Thank you for stating the following in the Acknowledgments Section of your manuscript: 

“The field work in Bangladesh was funded by the United States Agency of International Development (contract number US AID RE III 388-0070). This fieldwork began in July of 1997 and ended in August of 1997. All other costs have been paid from the personal savings of the authors. We are grateful to Marwan Said Abualrub, Ph.D. for reviewing our mathematical application of the Nernst equation.”

“The fieldwork in Bangladesh was funded by the United States Agency of International Development (USAID; contract number US AID RE III 388-0070; https://www.usaid.gov/). This fieldwork began in July of 1997 and ended in August of 1997. USAID paid Seth H. Frisbie’s (SHF) salary during these two months in 1997. All other costs have been paid from the personal savings of the authors. The funders had no role in study design, data collection and analysis, decision to publish, or preparation of the manuscript.”

RESPONSE TO JOURNAL REQUIREMENT 4: Done. The following three sentences were deleted from the Acknowledgments section: “The fieldwork in Bangladesh was funded by the United States Agency of International Development (contract number US AID RE III 388-0070). This fieldwork began in July of 1997 and ended in August of 1997. All other costs have been paid from the personal savings of the authors.”

The Acknowledgments section now says, “We thank Marwan Said Abualrub, Ph.D., and Amrutaa Vibho for reviewing our mathematical application of the Nernst equation.”

As requested, additional information has been added to the Financial Disclosure section; it now says, “The fieldwork in Bangladesh was funded by the United States Agency of International Development (USAID; contract number US AID RE III 388-0070; https://www.usaid.gov/). This fieldwork began in July of 1997 and ended in August of 1997. USAID is an international development agency that is funded by the United States government. USAID employed Seth H. Frisbie (SHF) and paid his salary during these two months in 1997. After August 1997, SHF received no specific funding for this work. Erika J. Mitchell (EJM) and Azizur R. Molla (ARM) received no specific funding for this work. No commercial companies funded the study or the authors.

All other costs have been paid from the personal savings of the authors. The funders had no role in study design, data collection and analysis, decision to publish, or preparation of the manuscript.”

JOURNAL REQUIREMENT 5: We note that Figures 1, 5, 6, 8, 11, 13, 15 in your submission contain [map/satellite] images which may be copyrighted. All PLOS content is published under the Creative Commons Attribution License (CC BY 4.0), which means that the manuscript, images, and Supporting Information files will be freely available online, and any third party is permitted to access, download, copy, distribute, and use these materials in any way, even commercially, with proper attribution. For these reasons, we cannot publish previously copyrighted maps or satellite images created using proprietary data, such as Google software (Google Maps, Street View, and Earth). For more information, see our copyright guidelines: http://journals.plos.org/plosone/s/licenses-and-copyright.

1.) You may seek permission from the original copyright holder of Figures 1, 5, 6, 8, 11, 13, 15 to publish the content specifically under the CC BY 4.0 license.

Please upload the completed Content Permission Form or other proof of granted permissions as an ““Other”“ file with your submission.

2.) If you are unable to obtain permission from the original copyright holder to publish these figures under the CC BY 4.0 license or if the copyright holder’s requirements are incompatible with the CC BY 4.0 license, please either i) remove the figure or ii) supply a replacement figure that complies with the CC BY 4.0 license. Please check copyright information on all replacement figures and update the figure caption with source information. If applicable, please specify in the figure caption text when a figure is similar but not identical to the original image and is therefore for illustrative purposes only.

RESPONSE TO JOURNAL REQUIREMENT 5: Done. We created these maps ourselves using ArcGIS® Online software by Esri®. The language recommended by Esri® for citing the use of ArcGIS® Online was taken from https://support.esri.com/en-us/knowledge-base/faq-what-is-the-correct-way-to-cite-an-arcgis-online-ba-000012040 and is included in the caption of each figure.

The basemap was published by United States Geological Survey and is in the public domain. The language recommended by the United States Geological Survey for citing this base map was taken from https://doi.org/10.3133/ofr97470H and this citation is included in the caption of each figure.

COMMENTS TO THE AUTHOR

COMMENT 1: Is the manuscript technically sound, and do the data support the conclusions?

Reviewer #1: Yes

Reviewer #2: Yes

RESPONSE 1: We agree.

COMMENT 2: Has the statistical analysis been performed appropriately and rigorously?

Reviewer #1: Yes

Reviewer #2: I Don’t Know

RESPONSE 2: In RESPONSE TO JOURNAL REQUIREMENT 3, we have significantly expanded the Supporting Information section of the manuscript from 1 file to 7 files as follows.

S1 Checklist. Inclusivity in global research questionnaire. (DOCX)

S2 File. Raw data, calculations, and independent statistical analyses. (XLSX)

S3 File. The R code that was used to analyze the arsenic concentration, dissolved oxygen concentration, oxidation-reduction potential, specific conductance, pH, and temperature data in S2 File.xlsx. (R)

S4 File. The derivation of a pH-dependent Nernst equation. (DOCX)

S5 File. Acid dissociation data for H3As(V)O4 and H3As(III)O3. (XLSX)

S6 File. The R code that was used to graph the acid dissociation data for H3As(V)O4 in S5 File.xlsx. (R)

S7 File. The R code that was used to graph the acid dissociation data for H3As(III)O3 in S5 File.xlsx. (R)

Our expanded list of supporting files includes independent statistical analyses of the entire data set in both R and Microsoft® Excel®. It also includes the R code for these analyses. Both R and Microsoft® Excel® give identical results. This way, readers can use either R or Microsoft® Excel® to evaluate our statistical analysis. The authors hope that this will make our work more accessible.

COMMENT 3: Have the authors made all data underlying the findings in their manuscript fully available?

Reviewer #1: Yes

Reviewer #2: No

RESPONSE 3: Again, in response to JOURNAL REQUIREMENT 3 and the previous comment, we have significantly expanded the Supporting Information section of the manuscript from 1 file to 7 files. All our data, including the raw data, two independent statistical analyses of the raw data, calculations of the dissociation of H3As(V)O4 and H3As(III)O3 as a function of pH, and the derivation and application of a pH-dependent Nernst equation, are submitted to comply with the PLOS Data Policy.

COMMENT 4: Is the manuscript presented in an intelligible fashion and written in standard English?

Reviewer #1: Yes

Reviewer #2: Yes

RESPONSE 4: We agree.

5. Review Comments to the Author

COMMENT 5: Reviewer #1: The paper by Frisbie et al. is an interesting paper on the connection between sea level rise and water quality beyond just salinization of coastal aquifers. It brings to the reader a known problem in Bangladesh but under a different light.

The paper requires a complete restructuring and rewriting of the sections. Moreover, figures should be combined given the similarities. It is also too long and the statistical analysis of the data cannot be related to the detailed geochemistry reported.

RESPONSE 5: Done. We agree, this paper has several novel aspects, including the realization that as annual flooding increases in Bangladesh due to sea level rise from climate change, this will increase the release of arsenic into drinking well water by reduction. To our knowledge, this paper is the first to show that as the area and duration of saltwater intrusion into Bangladesh’s aquifer increases from sea level rise, this will increase the release of arsenic into drinking well water by the salt effect.

As shown in our response to the comments below, we have restructured and shortened the manuscript, reduced the number of figures, and clarified our explanation of our statistical analysis. In particular, “The reported depth of the drinking water well” subsection was deleted, Figure 2 was deleted, and the discussion of the Benjamini and Hochberg’s False Discovery Rate in statistics was reworded (please see RESPONSES 15, 7, and 12, respectively).

The graphs all use arsenic concentration in micrograms per liter (µg/L) for the y-axis; however, they all have different units for the x-axis, so they cannot be combined into a single graph.

COMMENT 6:The manuscript needs to be re-written in a simpler structure including sections such as: Introduction, materials and methods reporting the sites and the statistical analysis and the analytical methods, a section on the geochemistry where the relationships between the salinity and the field data are highlighted to support the hypotheses, results and discussion with integrated figures (currently there are too many figures), and a conclusion which does not summarizes the paper but it really highlights the outcomes.

Below are few detailed comments.

RESPONSE 6: Done. In response to both reviewers’ detailed comments, the manuscript has been rewritten, and the overall structure has been simplified. The manuscript uses the format of Introduction, Materials and Methods, Results and Discussion, and Conclusion. The Materials and Methods discuss the sample locations, analytical methods, and statistical analysis. The two types of chemistry, oxidation-reduction and salt effect, are described in separate subsections of the Results and Discussion. All figures are cited in the text and located after their initial citations. The Conclusion has been edited to stress the outcomes.

COMMENT 7: Detailed comments

The introduction should be a single section starting from the overall problem of salinization of coastal aquifers and pollution or directly arsenic mobilization. Currently, it is too detailed regarding the problem of Bangladesh and it needs to be revised leaving detailed aspects of the site towards the end of the introduction.

RESPONSE 7: Done. The Introduction had six subsections; it now has three subsections. Keeping these subsections was done to define each topic’s content clearly, improve understanding, and allow readers to focus on the topics that interest them most; this is especially important in an environmental science paper whose readers have diverse interests and backgrounds.

The Introduction gives an overview of the problem of the salinization and pollution of coastal aquifers from flooding, briefly describes the problem of arsenic in Bangladesh, and introduces our study on arsenic mobilization.

As requested in COMMENTS 9 and 13, our detailed site description has been moved to the Materials and Methods section.

COMMENT 8: Lines 79 and 82: the number should be written in letters. This is repeated throughout the paper. Please correct.

RESPONSE 8: Done. All numbers used for counting that are less than or equal to 10 are now expressed as words, not as numbers. For example, “1 of the largest” now reads “one of the largest” (please see Former Line Number 87), and “These 2 processes” now reads “These two processes” (please see Former Line Number 79).

All numbers used to express a measurement are unchanged to reflect the precision of the measurement. For example, “8 meters” still reads as “8 meters” (please see Former Line Number 90).

COMMENT 9: Figure 1 should be moved in material and methods and not in the introduction. The map should also report more details about the region to help the reader to identify locations such as coast, cities, and boundaries.

RESPONSE 9: Done. As requested, Figure 1 was moved from the Introduction to the Materials and Methods.

Figure 1 and the other maps in this article clearly show Bangladesh’s coastline, national boundaries, elevation above sea level, and a scale in miles and kilometers. These maps used a basemap published public domain by the United States Geological Survey. Currently, these maps comply with PLOS ONE’s very appropriate and stringent licensing requirements (please see RESPONSE TO JOURNAL REQUIREMENT 5). No further edits to these maps were made to avoid potential licensing problems, minimize congestion, and maintain clarity.

COMMENT 10: Figure 2 can be broad to the SI document and just described with words.

RESPONSE 10: Done. This figure was deleted from the manuscript; however, the description of its significance remains in the text. Deleting this figure also supports this reviewer’s request to reduce the number of figures in the manuscript.

COMMENT 11: Line 219: I would instead write: “Our research aims at identifying the relationship between the chemical changes of the coastal aquifer due to sea level rise along the coast of Bangladesh and the increase release of arsenic into drinking water wells”.

RESPONSE 11: Done. Excellent suggestion. We edited your sentence and are using it as the topic sentence in this paragraph. It now reads, “Our research aims to identify the relationship between the chemical changes of the coastal aquifer due to sea level rise and the increased release of arsenic from sediments into Bangladesh’s drinking well water.”

COMMENT 12: Line 269: Why was the “Benjamini and Hochberg’s False Discovery Rate” method chosen and not another one? A short description of the method and a justification of the choice should be included.

RESPONSE 12: Done. We selected the Benjamini and Hochberg False Discovery Rate method because it is the most commonly used method for correcting for multiple comparisons More specifically, the 1995 paper by Benjamini and Hochberg on controlling false positives during multiple comparisons is one of the 25 most cited papers in statistics (Ryan and Woodall, 2005; https://www.tandfonline.com/doi/abs/10.1080/02664760500079373). As requested, the description of this method in the manuscript was expanded.

COMMENT 13: Section Mapping should be integrated with the description of the site.

RESPONSE 13: Done. The Mapping subsection is in the Materials and Methods section; it describes how our maps were made and the requirements for properly citing the use of the ArcGIS® Online software (please see response to JOURNAL REQUIREMENT 5). As requested, the Physical Environment subsection includes the site description and was moved to the Materials and Methods section (please see RESPONSE 7); it describes Bangladesh’s topology and other aspects of this region.

COMMENT 14: Line 330 and 374 should be a section named geochemistry where the fundamental of the chemistry of arsenic are explained to support all the arguments raised by the measurements. This should be combined with section titled: The Nernst equation and the release of arsenic.

RESPONSE 14: Done. We have collated the material about the fundamentals of the geochemistry of arsenic into the Results section. Our two main chemistry sections in the Results section deal with reduction and the salt effect. Each of these main sections is divided into subsections for clarity. Unfortunately, the distinction between main sections and subsections is shown only by the format of the headings since outline numbering is not permitted according to the journal’s style sheet. Since this distinction between levels may be overlooked by some readers, we have tried to emphasize the structure by choosing titles and subtitles with parallel structure according to level.

COMMENT 15: Line 897: I would bring up the relationship between conductivity and depth that are statistically significant and then that with oxygen.

RESPONSE 15: Done. In response to COMMENT 5, this subsection was deleted to shorten the manuscript. As stated in the original manuscript, this subsection was limited because “The following is an analysis of reported depths, not measured depths; as a result, it is subject to recall error and limits this aspect of our study.” If the authors must shorten the manuscript, we agreed it would be best to delete this subsection.

COMMENT 16: The conclusion section should just briefly state the problem and summarize the major outcomes of the paper. It is instead a combination of abstract and manuscript summary.

RESPONSE 16: Done. We have restructured this section as suggested.

Reviewer #2: The manuscript is about the “Sea level rise from climate change is expected to increase the release of arsenic into Bangladesh’s drinking well water by reduction and by the salt effect”

Some comments are suggested as below:

COMMENT 17: What was the sampling volume?

RESPONSE 17: Done, “350 milliliter (mL)” was added to the following sentence in the Drinking Well Water Sampling and Analyses subsection of the Materials and Methods section. “Subsamples were also collected for analysis in the laboratory directly into 350 milliliter (mL) polyethylene bottles and were not filtered.”

COMMENT 18: It seems that according to the measured concentration, the graphite furnace atomic absorption spectrophotometry (GFAA) is much more suitable.

RESPONSE 18: We agree that GFAAS would have been preferable; however, in 1997, GFAAS was very rare in Bangladesh and unavailable to USAID and our team. Therefore, we used the silver diethyldithiocarbamate spectrophotometric method, which was approved for the analysis of water and wastewater by the American Public Health Association (APHA), American Water Works Association (AWWA), and Water Environment Federation (WEF).

APHA, AWWA, WEF. Standard methods for the examination of water and wastewater. 19th ed. Washington, DC: American Public Health Association; 1995. parts 2000, 3000, 4000, 5000.

COMMENT 19: rom any method of sampling, measurement has been done, it must be mentioned the source.

RESPONSE 19: The referenced sample collection and analysis methods were approved for the analysis of water and wastewater by the APHA, AWWA, WEF (please see RESPONSE 18). In addition, the latitude and longitude of all sample locations were identified using the Global Positioning System (GPS). These GPS coordinates of the sample locations are in the Supplemental Information section (please see S2 File.xlsx).

COMMENT 20: Please send me the R codes written for this article and its results in text file so that I can analyze it.

RESPONSE 20: Done. Our expanded list of supporting files includes independent statistical analyses of the entire data set in both R and Microsoft® Excel®. It also includes the R code for these analyses. Both R and Microsoft® Excel® give identical results. This way, readers can use either R or Microsoft® Excel® to evaluate our statistical analysis. The authors hope that this will make our work more accessible (please see response to JOURNAL REQUIREMENT 3 and RESPONSE 2).

COMMENT 21: Summarize the relationship of different cases with arsenic auditors and compare with other articles.

RESPONSE 21: The results of our five regression equations can be used to predict the concentrations of arsenic in Bangladesh’s drinking well water as functions of DO, ORP, SC, pH, and T. However, our regression equations cannot be used to predict human health effects.

COMMENT 22: How did you reach the arsenic ions by changing the pH? More explanation is needed.

RESPONSE 22: Done. The captions of Figures 8 and 9 were expanded to explain that the concentrations of H3As(V)O4 and H3As(III)O3 species as a function of pH were calculated from equilibrium constants. In addition, the three files used to make these calculations and draw these figures were made available in the Supporting Information section. These files are S5 File.xlsx, S6 File.R, and S7 File.R.

COMMENT 23: The effects of arsenic in places where their levels are very high should be determined. Do you have any data in this field?

RESPONSE 23: Done. The primary concern of this paper is geochemistry, and our sampling was strictly limited to drinking water. This paper did not use human subjects. Thus, the paper does not contribute to further knowledge about the human health effects of exposure to arsenic in drinking well water. Nevertheless, Arsenic in Bangladesh’s Environment subsection in the Introduction does include a brief statement about the consequences for human health due to arsenic exposure.

COMMENT 24: Considering that the amount of arsenic in Bangladesh is high and even reaches more than 40 times the standard in some places, describe its effects in Bangladesh using other studies that exist.

RESPONSE 24: Done. Again, the primary concern of this paper is geochemistry, and our sampling was strictly limited to drinking water. This paper did not use human subjects. Thus, the paper does not contribute to further knowledge about the human health effects of exposure to arsenic in drinking well water. Nevertheless, Arsenic in Bangladesh’s Environment subsection in the Introduction does include a brief statement about the consequences for human health due to arsenic exposure.

6. PLOS authors have the option to publish the peer review history of their article (what does this mean?). If published, this will include your full peer review and any attached files.

Do you want your identity to be public for this peer review? For information about this choice, including consent withdrawal, please see our Privacy Policy.

Reviewer #1: No

Reviewer #2: No

COMMENT 25: While revising your submission, please upload your figure files to the Preflight Analysis and Conversion Engine (PACE) digital diagnostic tool, https://pacev2.apexcovantage.com/. PACE helps ensure that figures meet PLOS requirements. To use PACE, you must first register as a user. Registration is free. Then, login and navigate to the UPLOAD tab, where you will find detailed instructions on how to use the tool. If you encounter any issues or have any questions when using PACE, please email PLOS at figures@plos.org. Please note that Supporting Information files do not need this step.

RESPONSE 25: Done.

---

## [Decision Letter · Decision Letter 1]

17 Nov 2023

Sea level rise from climate change is expected to increase the release of arsenic into Bangladesh’s drinking well water by reduction and by the salt effect

PONE-D-23-21817R1

Dear Dr. Frisbie,

We’re pleased to inform you that your manuscript has been judged scientifically suitable for publication and will be formally accepted for publication once it meets all outstanding technical requirements.

Kind regards,

Venkatramanan Senapathi, Ph.D.

Academic Editor

PLOS ONE

Additional Editor Comments (optional):

The author has incorporated the reviewer comments into the paper, resulting in revisions that address the suggested improvements. Consequently, the manuscript is now deemed suitable for acceptance and publication.

Reviewers' comments:

Reviewer's Responses to Questions

**Comments to the Author**

1. If the authors have adequately addressed your comments raised in a previous round of review and you feel that this manuscript is now acceptable for publication, you may indicate that here to bypass the “Comments to the Author” section, enter your conflict of interest statement in the “Confidential to Editor” section, and submit your "Accept" recommendation.

Reviewer #1: All comments have been addressed

Reviewer #2: All comments have been addressed

2. Is the manuscript technically sound, and do the data support the conclusions?

Reviewer #1: Yes

Reviewer #2: Yes

3. Has the statistical analysis been performed appropriately and rigorously? 

Reviewer #1: Yes

Reviewer #2: Yes

4. Have the authors made all data underlying the findings in their manuscript fully available?

Reviewer #1: Yes

Reviewer #2: Yes

5. Is the manuscript presented in an intelligible fashion and written in standard English?

Reviewer #1: Yes

Reviewer #2: Yes

6. Review Comments to the Author

Reviewer #1: The authors have addressed all the comments in my review and the paper can be published in the present form.

Reviewer #2: I read the manuscript, and all comments are improved that. Due to this issue I recommend to publication of this article in journal.

7. PLOS authors have the option to publish the peer review history of their article (what does this mean?). If published, this will include your full peer review and any attached files.

Reviewer #1: No

Reviewer #2: No

---

## [Editor Report · Acceptance letter]

6 Dec 2023

PONE-D-23-21817R1 

Sea level rise from climate change is expected to increase the release of arsenic into Bangladesh’s drinking well water by reduction and by the salt effect 

Dear Dr. Frisbie:

I'm pleased to inform you that your manuscript has been deemed suitable for publication in PLOS ONE. Congratulations! Your manuscript is now with our production department. 

Kind regards, 

on behalf of

Dr. Venkatramanan Senapathi 

Academic Editor

PLOS ONE